# TRANSFER BOUND OF GRAPH CONVOLUTIONAL NETWORKS ACROSS ARBITRARY SPARSITY

## ABSTRACT

Size transfer in Graph Convolutional Networks (GCNs) is a common treatment to mitigate the high cost of training on large graphs, by transferring the model trained on randomly sampled smaller graphs. However, the theoretical guarantee of such transfer has only been proved in previous studies for random graphs with restricted sparsity. In practice, downsampled real-world graphs may exhibit multiple sparsity regimes. To fully understand the theoretical performance across sparsity, we establish the GCN transferability bound by introducing Stretched Graphon Convolutional Networks (SWNNs) based on the recent generalized graphon model. The bound decomposes into error components arising from expected edge density and graph size, which jointly determine the sparsity. Experiments on real-world networks validate our theoretical findings.

## 1 INTRODUCTION

Graph convolutional networks (GCNs) (Defferrard et al., 2016; Kipf & Welling, 2017) have achieved great success in real-world tasks containing network structures, such as social networks (Sharma et al., 2024), knowledge graphs (Wang et al., 2019), and citation networks (Zhang et al., 2019; Xia et al., 2023). However, training GCNs on large graphs costs substantial computational resources (Liu et al., 2021; Zhang et al., 2023). Size transfer is usually adopted to address this issue, by which GCNs are trained on downsampled smaller graphs and transferred to larger graphs (Chen et al., 2018; Chiang et al., 2019; Yao & Li, 2021). In this process, nodes and edges are randomly selected from the original graph, producing subgraphs that may have multiple sparsity levels. Here, sparsity characterizes the decreasing dependence of expected edge density on graph size.

Although size transfer has demonstrated empirical effectiveness, further theoretical analysis and guarantees are demanding (Cordonnier, 2024; Morris et al., 2024; Levin et al., 2025). Most analyses of GCN transferability rely on graph similarity, typically formalized through graph models. Based on these models, generated graph sequences share similar topologies and converge to model-related limits as the graph size increases to infinity (Janson, 2010; Lovász, 2012; Backhausz & Szegedy, 2022; Ji et al., 2024). In line with this sequence convergence, GCNs also converge to a corresponding limit, where the convergence errors directly bound the transferability error. Some existing works analyze GCNs using the classical graphon model (Ruiz et al., 2020; Magner et al., 2022; Ruiz et al., 2023; Maskey et al., 2023), where the generated graphs are dense (Orbanz & Roy, 2015; Borgs et al., 2018) with expected edge density $\Theta(1)$.

Real-world networks are typically sparse with decreasing edge densities and may exhibit multiple sparsity levels across downsampled graphs, making GCN analysis on such graphs particularly challenging. Some studies of GCNs on sparse graphs do not generate concrete graphs (Levie et al., 2021; Le & Jegelka, 2023). Other works guarantee convergence or transferability only under restricted sparsity regimes, such as $\Theta(\log n/n)$ or lower for graph size $n$ (Keriven et al., 2020; 2021; Wang et al., 2023; 2024). Consequently, when considering transfer between graphs across different sparsity levels, including higher sparsity, existing results do not address this case. To broaden the transferability analysis of sparse GCN, it is necessary to establish a theoretical guarantee that holds across arbitrary sparsity levels.

In this paper, we address the challenge of sparse GCN transfer across multiple sparsity, inspired by the generalized graphon model (Borgs et al., 2018; 2019; Ji et al., 2024). Our approach guarantees transferability across arbitrary sparsity and provides explicit transfer bounds. The generalized

graphon model generates sparse graphs through double-sampling and employs stretching to ensure a double-convergent non-zero limit. Stretching amplifies the structures of sparse graphs to vanishing effects caused by sparsity (Borgs et al., 2018; 2019; Ji et al., 2024). Building upon these, we introduce stretched graphon convolutional networks (SWNNs) as the meaningful limit of GCNs over sparse graph sequences, and derive the transfer error bound.

Our contributions to this work are as follows:

- We introduce SWNN, the stretched graphon convolutional network, as the non-zero limit of GCNs over sparse graph sequences across multiple sparsity, providing a meaningful reference for convergence and transferability analysis.

- We prove that GCN transferability holds across arbitrary sparsity, and establish an explicit upper bound on the transferability error. The bound composed of convergence errors is derived by graph sampling errors, which are analyzed at the two phases of the double-sampling procedure, and combined into a unified result.

## 2  RELATED WORK

The widespread applicability of GCNs originates from their ability to propagate and aggregate information on irregular graph structures (Defferrard et al., 2016; Kipf & Welling, 2017; Zhang et al., 2019). However, a theoretical question arises: how to quantify the difference between different graphs? This is generally achieved by modeling random graphs and establishing convergence within models (Cordonnier, 2024; Morris et al., 2024; Levin et al., 2025), such as the classical graphon model for dense case (Lovász, 2012; Ruiz et al., 2021a), and the generalized graphon model for sparse case Borgs et al. (2018; 2019); Ji et al. (2024).

**Transfer in Dense Graphs**. Within dense graph sequences employed for analyzing GCN transferability, the classical graphon is a widely adopted framework (Ruiz et al., 2020; Magner et al., 2022; Cai & Wang, 2022; Ruiz et al., 2023; Maskey et al., 2023; Cerviño et al., 2023; Velasco et al., 2024). Measured by cut norm and cut distance, the classical graphon ($W : [0,1]^2 \rightarrow [0,1]$) provides a functional representation of graph topology. It serves as a generative model for dense graph sequences and the limit of such sequences (Janson, 2010; Lovász, 2012). For a given graphon $W$, the expected edge density remains constant $\|W\|_1$, making edges scale at a dense rate $n^2\|W\|_1$. Ruiz et al. (2021a) extend the graphon model by incorporating graphon signals ($X : [0,1] \rightarrow \mathbb{R}$), allowing for a joint sampling of graph topology and node features, and further establish graphon signal processing framework. In their subsequent work (Ruiz et al., 2020; 2023), the convergence of graph sequences is extended to GCN sequences, then the GCN transferability is analyzed. Furthermore, Maskey et al. (2023) broaden the classical graphon to value-unbounded graphon and analyze the GCN with various convolutional forms. Based on the classical graphon, the graph classification capability (Magner et al., 2022) and the transference training efficiency (Cerviño et al., 2023) of GCNs are analyzed, the conclusions on GCNs are extended to other graph neural networks (Cai & Wang, 2022; Velasco et al., 2024).

**Transfer in Sparse Graphs**. Since real-world networks are typically sparse with decreasing edge densities, sparse graph models are adopted in analyzing the transferability of GCNs (Keriven et al., 2020; 2021; Wang et al., 2023; 2024; Ruiz et al., 2024). In Keriven et al. (2020), sparse graph sequences are constructed by multiplying the classical graphon with a decreasing sparsity factor $\alpha_n$. As the graph size grows, the graphon tends to zero, resulting in vanishing edge densities. Instead of the zero limit, they introduce a normalized Laplacian operator divided by the sparse degrees, yielding a relatively sparse convergence conclusion. Keriven et al. (2021) normalize adjacency matrix by increasing edge weights reversely proportional to sparsity ($\alpha_n^{-1}\text{Ber}(\alpha_n W)$), ensuring that expected edge weights remain stable. Although sparsity weakens connectivity, their enhanced weights reinforce the propagation along existing edges. Similarly, Wang et al. (2023; 2024) construct relatively sparse geometric graphs with increasing weights inversely proportional to sparsity, and study the GCN convergence to manifold neural networks. In general, these works analyze convergence or transferability under models with restricted sparsity levels, such as relative sparsity or lower. However, transfer between graphs across different sparsity levels, including higher sparsity, remains unaddressed. Based on generalized graphon, Ruiz et al. (2024) construct sparse graphs without sparsity restriction, but their GCN spectral convergence error does not vanish and approaches a con-

stant. These highlight the need for a transferability result on sparse random graphs across arbitrary sparsity levels. Other modeling approaches obtain graph operators directly through discretization on geometric spaces or continuous operators, without generating explicit random graphs (Levie et al., 2021; Le & Jegelka, 2023). Since no concrete graphs are produced, these approaches cannot, as we do, physically decompose the transfer error into components attributable to graph size and edge density. Distinct from works on topology inference (Rey et al., 2023) or training parameter convergence (Alimohammadi et al., 2025), our research targets transferability in sparse graphs. We specifically quantify the output discrepancy of GCNs across varying sparsity levels, rather than analyzing the learning process

## 3 THE SPARSE RANDOM GRAPH MODEL

In this paper, we consider a sparse random graph model based on the generalized graphon framework (Borgs et al., 2018; 2019; Ji et al., 2024). This line of work is closely related to the graphex framework (Borgs et al., 2021), which provides a comprehensive limit object capturing structural properties of sparse graphs. Given that the convolutional operator is the key to GCNs, we adopt the signal processing framework from Ji et al. (2024), which is precisely tailored to analyze the limit of sparse graph sequences.

In this model (Ji et al., 2024), sparse graphs are generated through a double-sampling procedure: first, a sparse classical graphon sequence $\{W_m\}$ is sampled, and then sparse graphs $\{G_{m,n}\}$ are sampled from this sequence. This construction enables the generation of graph subsequences with arbitrary sparsity (see Fig. 2),

$$\text{Sparsity(Graph size)} = \text{Expected edge density}. \tag{1}$$

Here, edge density is a topological property defined as the ratio of the realized number of edges $e(G_n)$ to the total possible edges $n^2/2$, and sparsity refers to the decay function of the expected edge density as graph size $n$ increases (Keriven et al., 2020; Le & Jegelka, 2023; Wang et al., 2023).

**Definition 1** (Generalized graphon and signal). *A generalized graphon $W_{\mathbb{R}_+}(u, v)$ is a symmetric function: $\mathbb{R}_+^2 \to [0, 1]$. A generalized graphon signal $X_{\mathbb{R}_+}(u)$ is a function: $\mathbb{R}_+ \to \mathbb{R}$.*

We require both $W_{\mathbb{R}_+}$ and $X_{\mathbb{R}_+}$ to be $L^1$-integrable and bounded, ensuring that stretching and convolution operations are well defined. The topological variation between generalized graphons is measured using the extended cut norm and cut distance (Borgs et al., 2018; 2019). By extending the integration domain to $\mathbb{R}_+^2$, they are defined:

$$\delta_{\square}(W_{\mathbb{R}_+,1}, W_{\mathbb{R}_+,2}) = \inf_{\phi_1, \phi_2} \sup_{U,V \subseteq \mathbb{R}_+} \left| \int_{U \times V} (W_{\mathbb{R}_+,1}^{\phi_1} - W_{\mathbb{R}_+,2}^{\phi_2}) du dv \right|, \tag{2}$$

where $W_{\mathbb{R}_+,1}, W_{\mathbb{R}_+,2}$ are two generalized graphons; the sup is taken over measurable subsets $U, V \subseteq \mathbb{R}_+$, and the inf is taken over measure preserving bijection $\phi_1, \phi_2$. When applied to graphons induced by finite graphs, these bijections correspond to disregarding node ordering.

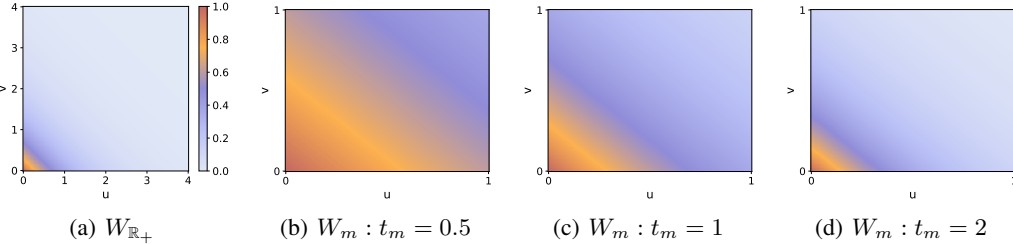

(a) $W_{\mathbb{R}_+}$      (b) $W_m : t_m = 0.5$      (c) $W_m : t_m = 1$      (d) $W_m : t_m = 2$

Figure 1: Example of sparse graphon sequence: (a) $W_{\mathbb{R}_+}$ denotes the original generalized graphon, (b)-(d) illustrate $W_m$, obtained by truncating the domain $[0, t_m]^2$ of $W_{\mathbb{R}_+}$ and rescaling it to $[0, 1]^2$.

### 3.1 THE SPARSE GRAPHON SEQUENCE

The sparse graphon sequence is constructed through truncation and rescaling. At stage $m \in \{1, 2, 3, \dots\}$, $W'_m$ is obtained by truncating $W_{\mathbb{R}_+}$ to the expanding square region $[0, t_m]^2$, setting the remaining area to zero (Ji et al., 2024). The truncated generalized graphon $W'_m$ is stage-expanding and eventually covers the entire $W_{\mathbb{R}_+}$. Then, $W'_m$ is rescaled to the standard domain $[0, 1]^2$ to obtain the sparse classical graphon $W_m$:

$$\textbf{Truncate: } W'_m(u, v) = \begin{cases} W_{\mathbb{R}_+}(u, v), & \text{where } u, v \in [0, t_m], \\ 0, & \text{others.} \end{cases} \tag{3}$$

$$\textbf{Rescale: } W_m(u, v) = W'_m(t_m u, t_m v), \quad \text{where } u, v \in [0, 1].$$

Similarly, the graphon signal $X_m$ is obtained by truncating $X_{\mathbb{R}_+}$ to the interval $[0, t_m]$ and subsequently rescaling it to $[0, 1]$: $X_m(u) = X'_m(t_m u) = X_{\mathbb{R}_+}(t_m u)$. Since $W_{\mathbb{R}_+}$ is required to be integrable over $\mathbb{R}_+^2$, the $L^1$-norm of the classical graphon sequence $\{W_m\}$ tends to zero at the rate of $\Theta(1/t_m^2)$, indicating sparsity, as illustrated in Figure 1.

### 3.2 THE GRAPH SEQUENCE SAMPLED FROM SPARSE GRAPHON

Based on the sparse graphon and signal $(W_m, X_m)$, a random graph $G_{m,n}$ is sampled following the classical graphon sampling procedure (Lovász, 2012; Ruiz et al., 2021a; Ji et al., 2024). Latent features for $n$ nodes, $\{u_1, u_2, \dots, u_n\}$, are sampled i.i.d. from a uniform distributions on $[0, 1]$. Node features are then obtained as $\boldsymbol{x}_{m,n}(i) = X_m(u_i)$, and edge probabilities are given by $\mathbf{P}_{m,n}(i, j) = p_{ij} = W_m(u_i, u_j)$. Finally, edges are sampled via Bernoulli distributions:

$$\mathbf{S}_{m,n}(i, j) \sim \text{Ber}(\mathbf{P}_{m,n}(i, j)), \quad 1 \le i \le j \le n. \tag{4}$$

In the overall sparse graph sequence $\{\{G_{m_1,n}\}, \{G_{m_2,n}\}, \dots\}$, each stage $m_l$ can generate graphs of varying sizes $\{G_{m_l,n}\}$ ($l \in \{1, 2, \dots\}$). By selecting specific graphs $\{G_{m_1,n_1}, G_{m_2,n_2}, \dots\}$ from different stages, one can construct graph subsequences with arbitrary sparsity, as shown in Fig. 2. For example, to realize a sparsity of $C/n$ with respect to graph size $n_l$, one may choose the corresponding $W_m$ such that $\|W_{m_l}\|_1 = C/n_l$.

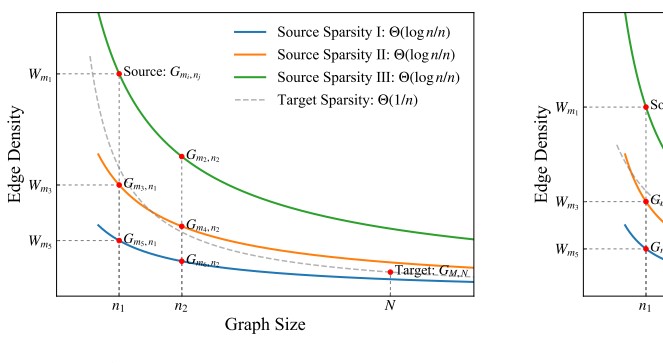

(a) $\Theta(\frac{\log n}{n})$ family of sparsity functions.   (b) $\Theta(\frac{1}{n})$ family of sparsity functions.

Figure 2: Graph subsequences with multiple sparsity, such as $\Theta(\frac{\log n}{n})$ in (a) and $\Theta(\frac{1}{n})$ in (b). Transfer errors are measured between the source graphs and target graph, while the target graph $G_{M,N}$ may lie beyond the sparsity line of source graphs $\{G_{m,n}\}$.

In real-world downsampling scenarios, sampling sparse subgraphs of size $n$ can produce multiple edge densities, leading to diverse sparsity levels. Previous works (Keriven et al., 2020; 2021; Wang et al., 2023; 2024) focus on models with restricted sparsity, corresponding to a single sparsity curve in Fig. 2, and guarantee transferability only for sparsity of $\Theta(\log n/n)$ or lower. However, the target graph may fall outside the sparsity regime of the sampled subgraphs, necessitating transfer analysis across different sparsity levels, as illustrated in Fig. 2.

To study the convergence of the graph sequence $\{G_{m,n}\}$, the adjacency matrix and node features are transformed into functional forms (Ruiz et al., 2020; 2021a; 2023; Maskey et al., 2023). The classical induced graphon and signal are defined as

$$\overline{W}_{m,n}(u \in \mathbb{I}_i, v \in \mathbb{I}_j) = \mathbf{S}_{m,n}(i,j), \quad \overline{X}_{m,n}(u \in \mathbb{I}_i) = \boldsymbol{x}_{m,n}(i), \tag{5}$$

where $\{\mathbb{I}_i\}_{i=1}^n$ are $n$ equal partitions of $[0,1]$. This inducing procedure enables comparisons across graphs of different sizes and between graphs and the underlying graphon.

### 3.3 CONVERGENCE UNDER THE STRETCHED FORM

For sparse graph sequences, the classical induced graphon $\{\{\overline{W}_{m_1,n}\}, \{\overline{W}_{m_2,n}\}, \dots\}$ converges to a trivial zero limit, rendering direct comparison with the limit meaningless. This issue is resolved via stretching (Borgs et al., 2018; 2019; Ji et al., 2024), which scales the graphon in proportion to its sparsity.

**Definition 2** (Stretched graphon and signal). *The stretched forms of generalized graphon and signal are defined as* $W_{\mathbb{R}_+}^{\mathfrak{s}}(u,v) = W_{\mathbb{R}_+}(s_R u, s_R v), X_{\mathbb{R}_+}^{\mathfrak{s}}(u) = X_{\mathbb{R}_+}(s_R u),$ *where* $s_R = \|W_{\mathbb{R}_+}\|_1^{1/2}$ *is the stretching coefficient.*

Stretching fixes the $L^1$-norm of the graphon to one, thereby amplifying structural patterns according to sparsity and counteracting vanishing effects. For a classical graphon $W_m$ and induced graphon $\overline{W}_{m,n}$, stretching is applied with zero-padding to accommodate domain differences (Borgs et al., 2018; 2019; Ji et al., 2024):

$$
\textbf{Stretched Graph: } \overline{W}_{m,n}^{\mathfrak{s}}(u,v) = \begin{cases} \overline{W}_{m,n}\left(s_{m,n}u, s_{m,n}v\right), & \text{for } (u,v) \in [0, \frac{1}{s_{m,n}}]^2, \\ 0, & \text{others,} \end{cases}
$$

$$
\textbf{Stretched Graph Signal: } \overline{X}_{m,n}^{\mathfrak{s}}(u) = \begin{cases} \overline{X}_{m,n}\left(s_{m,n}u\right), & \text{for } u \in [0, \frac{1}{s_{m,n}}], \\ 0, & \text{others,} \end{cases}
\tag{6}
$$

where $s_{m,n} = \|\overline{W}_{m,n}\|_1^{1/2}$ is the stretching coefficient that reflects the graph's edge density. This operation amplifies structural characteristics in proportion to sparsity, compensating for the vanishing effect observed in sparse cases (Figure 3).

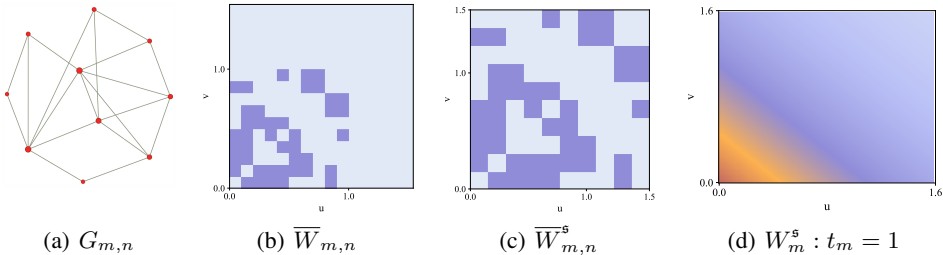

(a) $G_{m,n}$      (b) $\overline{W}_{m,n}$      (c) $\overline{W}_{m,n}^{\mathfrak{s}}$      (d) $W_m^{\mathfrak{s}} : t_m = 1$

Figure 3: Example of stretching: $G_{m,n}$ is a random graph sampled from $W_m$, its adjacency matrix is transformed to corresponding classical form (b) and stretched form (c), the latter converges to stretched graphon (d).

For a graph sequence $\{G_{m,n}\}$ generated by $W_m$, the stretched induced graphon $\overline{W}_{m,n}^{\mathfrak{s}}$ converges to $W_m^{\mathfrak{s}}$ under the generalized cut distance $\delta_\square$; For the sparse graphon $W_m$ truncated from $W_{\mathbb{R}_+}$, its stretched form converges to the non-zero limit $W_{\mathbb{R}_+}^{\mathfrak{s}}$ (Ji et al., 2024):

$$\lim_{m\to\infty}\lim_{n\to\infty}\overline{W}_{m,n}^{\mathfrak{s}} = \lim_{m\to\infty}W_m^{\mathfrak{s}} = W_{\mathbb{R}_+}^{\mathfrak{s}}. \tag{7}$$

Based on the double-sampling procedure, multiple graph subsequences with arbitrary sparsity can be constructed (see in Sec. 3.2). Although subgraphs $\{G_{m,n_l}|m\}$ with the same size $n_l$ but different expected edge densities are allowed under arbitrary sparsity, each $G_{m,n_l}$ is expected to approach its corresponding $W_m$. Convergence of these subsequences is guaranteed by the double convergence: at each stage $m$, graphs in a subsequence converge to the corresponding $W_m$, and the convergence of $W_m$ itself ensures overall convergence of the subsequences.

# 4 STRETCHED GRAPHON CONVOLUTIONAL NETWORKS

In this section, we introduce SWNNs, which serve both as the functional forms of GCNs and as their limit when applied to the generalized graphon and signal $(W_{\mathbb{R}_+}, X_{\mathbb{R}_+})$. Similar to the graph convolutional operator (Segarra et al., 2017; Du et al., 2018; Gama et al., 2019) in Appendix F.1, a graphon function can be regarded as an infinitely dense matrix with infinitely many nodes. The stretched graphon convolutional operator is conducted with integration (Ji et al., 2024),

$$h(W^{\mathfrak{s}})X^{\mathfrak{s}} = \sum_{r=0}^{R} h_r T_{W^{\mathfrak{s}}}^{(r)} X^{\mathfrak{s}},$$

$$\text{for } r \geq 1, \ T_{W^{\mathfrak{s}}}^{(r)} X^{\mathfrak{s}} = \int_{\mathbb{R}_+} W^{\mathfrak{s}} \left( T_{W^{\mathfrak{s}}}^{(r-1)} X^{\mathfrak{s}} \right) du. \tag{8}$$

The generalized graphon adopts the same convolutional form, since it is defined on $\mathbb{R}_+^2$ and subsumes the stretched graphon as a special case. Because convergence of sparse graph sequences is analyzed under the stretched setting, we focus primarily on the stretched form.

Let $\Phi(W^{\mathfrak{s}}, X^{\mathfrak{s}}, \mathcal{H})$ denote the stretched graphon convolutional network (SWNN), which follows the same framework and learnable parameters $\mathcal{H}$ as the GCNs (Ruiz et al., 2020; 2021b; 2023) in Appendix F.1, with one-dimensional input features for simplicity. At the $l$-th layer, the input and output feature dimensions are $F^{l-1}$ and $F^l$, with parameters $\mathcal{H}^l \in \mathbb{R}^{F^{l-1} \times F^l}$. The channel-wise convolutional processes of SWNNs parallel that of GCNs:

$$X_{f_l}^{\mathfrak{s}} = \sigma \left( \sum_{f_{l-1}=1}^{F_{l-1}} h_{f_{l-1}, f_l}(W^{\mathfrak{s}}) X_{f_{l-1}}^{\mathfrak{s}} \right), \tag{9}$$

where $\sigma(\cdot)$ denotes the activation function, and $h_{f_{l-1}, f_l}(\cdot) = \mathcal{H}^l(f_{l-1}, f_l)h(\cdot)$ contains the learnable parameters mapping the $f_{l-1}$-th input feature to the $f_l$-th output feature. These parameters are fixed when analyzing convergence and transferability. Notably, in SWNNs each layer's output $X_{f_l}^{\mathfrak{s}}$ is obtained recursively, rather than being defined explicitly in Definition 2, and thus it does not directly correspond to a particular $X_{f_l}$.

# 5 MAIN RESULT

In the following sections, we first present our main result of transferability. Then we analyze in detail the convergence of the sparse graphon sequence and the random graph sequence at each stage, which together underpin our main result.

The following assumptions are considered:
**AS1.** The generalized graphon $W_{\mathbb{R}_+}$ belongs to the Schwartz space. That is, for any $k > 0$, there exits a constant $C_k$ such that $W_{\mathbb{R}_+}(u, v) \leq C_k(u+1)^{-k}(v+1)^{-k}$. Thus, it is $A_W$-Lipschitz, i.e. $|W_{\mathbb{R}_+}(u_1, v_1) - W_{\mathbb{R}_+}(u_2, v_2)| \leq A_W(|u_1 - u_2| + |v_1 - v_2|)$.
**AS2.** The generalized signal $X_{\mathbb{R}_+}$ is $A_X$-Lipschitz, i.e. $|X_{\mathbb{R}_+}(u_1) - X_{\mathbb{R}_+}(u_2)| \leq A_X|u_1 - u_2|$.
**AS3.** The activation function $\sigma(\cdot)$ is normalized-Lipschitz, i.e. $|\sigma(x_1) - \sigma(x_2)| \leq |x_1 - x_2|$, and $\sigma(0) = 0$.

In AS1, we characterize the sparsity of $W_{\mathbb{R}_+}$ by requiring it to be rapidly decreasing. For most node pairs, the connection probability is small and generally decreases as the latent feature distance increases. In AS2, since sparsity is related to the graph topology, the generalized signal is only required to be Lipschitz continuous. In AS3, the assumption on activation functions holds for most commonly used functions, such as ReLU and Tanh.

## 5.1 TRANSFERABILITY OF SWNNS

In this section, we show that for two distinct stretched graphons, the output discrepancy of an SWNN can be bounded by the differences in the corresponding operators and input signals. This result directly leads to the convergence theorem of SWNNs (Theorem 2) and further establishes the corresponding transferability bound.

**Lemma 1** (Difference of SWNN Outputs). *Consider the L-layer SWNN with learned parameters $\mathcal{H}$, where $F_0 = F_1 = 1$, applied to stretched graphons with signals $(W_1^{\mathfrak{s}}, X_1^{\mathfrak{s}})$ and $(W_2^{\mathfrak{s}}, X_2^{\mathfrak{s}})$, then*

$$\left\| \Phi(W_1^{\mathfrak{s}}, X_1^{\mathfrak{s}}, \mathcal{H}) - \Phi(W_2^{\mathfrak{s}}, X_2^{\mathfrak{s}}, \mathcal{H}) \right\|_2 \leq C(\mathcal{H}) \left( \| W_1^{\mathfrak{s}} - W_2^{\mathfrak{s}} \|_{2,2} \| X_1^{\mathfrak{s}} \|_2 + \| X_1^{\mathfrak{s}} - X_2^{\mathfrak{s}} \|_2 \right), \quad (10)$$

*where $C(\mathcal{H})$ is a constant depending on $\mathcal{H}$ (proof in Appendix F.3).*

Now consider the source graph $G_{m,n}$ and target graph $G_{M,N}$, belonging to different subsequences with different sparsity, $\{G_{m_1,n_1}, G_{m_2,n_2}, \dots\}$ with Sparsity$_1$ and $\{G_{M_1,N_1}, G_{M_2,N_2}, \dots\}$ with Sparsity$_2$. These subsequences are constructed by selecting graphs according to the desired sparsity, as described in Sec. 3.2.

**Theorem 1** (Transferability of SWNNs). *Consider an L-layer SWNN with learned parameters $\mathcal{H}$, denoted by $\Phi(W^{\mathfrak{s}}, X^{\mathfrak{s}}, \mathcal{H})$, where $F_0 = F_1 = 1$ for simplicity. Under assumptions from AS1 to AS3, and conditions on truncation length $t_m, t_M$ and graph size $n, N$ in Lemma 2, Lemma 3, and Lemma 4, then for any $k' > 0$, it holds that*

$$\left\| \Phi(\overline{W}_{m,n}^{\mathfrak{s}}, \overline{X}_{m,n}^{\mathfrak{s}}, \mathcal{H}) - \Phi(\overline{W}_{M,N}^{\mathfrak{s}}, \overline{X}_{M,N}^{\mathfrak{s}}, \mathcal{H}) \right\|_2$$

$$\leq C(\mathcal{H}) \left( C_{\mathbb{R}_+} \left( \frac{1}{t_m^{k'}} + \frac{1}{t_M^{k'}} \right) + \frac{L_X^2(t_m) + L_X^2(t_M)}{\| W_{\mathbb{R}_+} \|_1^{1/4}} \right) + \mathcal{E}(n) + \mathcal{E}(N), \quad (11)$$

$$\text{where } \mathcal{E}(n) = C(\mathcal{H}) \left( \frac{C_{m,1}}{n^{1/2}} + \frac{C_{m,2}}{n^{1/4}} \right), \ \mathcal{E}(N) = C(\mathcal{H}) \left( \frac{C_{M,1}}{N^{1/2}} + \frac{C_{M,2}}{N^{1/4}} \right),$$

*with probability at least $(1 - \epsilon_{u,1})(1 - \epsilon_{b,1})(1 - \epsilon_{u,2})(1 - \epsilon_{b,2})$. Here, $C(\mathcal{H})$ depends on $\mathcal{H}$, $C_{\mathbb{R}_+}$ depends on $W_{\mathbb{R}_+}$ and $X_{\mathbb{R}_+}$, $L_X^2(\cdot)$ is the tail integral of $X_{\mathbb{R}_+}$, $C_{m,1}, C_{m,2}, C_{M,1}, C_{M,2}$ are constants about $W_m, X_m$ and $W_M, X_M$ (proof in Appendix F.4).*

Transferability across arbitrary sparsity is influenced by graph sizes $n, N$ and the expected edge densities at these sizes: Sparsity$_1(n)$, Sparsity$_2(N)$ in Eq.(1), which is given by $\| W_m \|_1 \approx \| W_{\mathbb{R}_+} \|_1 / t_m^2$ at $n$ and $\| W_M \|_1 \approx \| W_{\mathbb{R}_+} \|_1 / t_M^2$ at $N$ as shown in Fig. 2. For fixed graph sizes, the double-convergence property ensured that the error components related to graph sizes are of smaller order than those driven by edge densities; hence, increasing $t_m, t_M$ (i.e., reducing expected edge densities: $\| W_m \|_1, \| W_M \|_1$) yields stronger transferability. Conversely, for fixed expected edge density, enlarging graph size $n, N$ further improves transferability. Overall, the expected edge density serves as the dominant factor, while graph size provides an additional improvement, consistent with the double-convergence framework.

Here, we guarantee transferability across graph subsequences with arbitrary sparsity, while prior works (Keriven et al., 2020; 2021; Wang et al., 2023; 2024) analyze GCN only under restricted sparsity settings. In our framework, transfer errors vanish as both $t_m, t_M$ and $n, N$ increase to infinity, while the convergence errors in Ruiz et al. (2024) converge to constants. Building on this bound, we further analyze how edge density and graph size affect transferability, a perspective that Levie et al. (2021); Le & Jegelka (2023) cannot provide since they do not generate explicit graphs and thus cannot capture topological properties such as edge density.

The above GCN transfer error is basically derived from the convergence error in Theorem 2, measured against the SWNN limit. As established in Lemma 1, the discrepancy between SWNNs is bounded by differences in operators and signals. Therefore, in the following sections, we analyze these differences according to the double-sampling procedure in Lemma 2 and Lemma 3, which corresponds to double-convergence framework.

## 5.2 Convergence of Sparse Graphon Sequence

The convergence of a sparse graphon sequence is straightforward. Since $W_m$ and $X_m$ are obtained via truncation and rescaling, their stretched forms coincide with $W_m^{'\mathfrak{s}}$ and $X_m^{'\mathfrak{s}}$. As the truncation width $t_m$ increases, $W_m$ and $X_m$ converge to the original generalized graphon and signal.

**Lemma 2** (Convergence of Sparse Graphon Sequence). *Let $(W_{\mathbb{R}_+}, X_{\mathbb{R}_+})$ satisfy AS1 and AS2, and let $(W_m, X_m)$ be the sparse graphon and signal generated from them. Assume $t_m$ is sufficiently large such that $\| W_m' \|_1 \geq \| W_{\mathbb{R}_+} \|_1 / 2$. Then, for any $k' > 0$,*

$$\left\| W_{\mathbb{R}_+}^{\mathfrak{s}} - W_m^{\mathfrak{s}} \right\|_{2,2} \leq \frac{C_{k',W}}{t_m^{k'}}, \quad \left\| X_{\mathbb{R}_+}^{\mathfrak{s}} - X_m^{\mathfrak{s}} \right\|_2 \leq \frac{C_{k',X}}{t_m^{k'}} + \frac{L_X^2(t_m)}{\| W_{\mathbb{R}_+} \|_1^{1/4}}, \quad (12)$$

*where $L_X^2(t_m) = \left\| X_{\mathbb{R}_+} \right\|_{L^2([t_m,+\infty))}$ is the tail integral of $X_{\mathbb{R}_+}$ on $[t_m, +\infty)$, constants $C_{k',W}$, $C_{k',X}$ depend on $k'$, $W_{\mathbb{R}_+}$, and $X_{\mathbb{R}_+}$ (proof in Appendix D).*

This result not only establishes the convergence of sparse graphons $W_m$ but also provides explicit error bounds. Under the rapid decay condition on $W_{\mathbb{R}_+}$, $W_m$ converges at an accelerated rate. In contrast, the convergence of $X_m$ is slower, governed by the tail integral over the expanding domain, and influenced by $W_m$ through the stretching.

## 5.3 Convergence of Random Graph Sequence

At each stage $m$, the random graph sequence $\{G_{m,n}\}$ has its associated limit $W_m$. This convergence holds in both the classical form and the stretched form. Moreover, convergence in the stretched form is implied by convergence in the classical form. Their notation is summarized in Table 1.

Table 1: Notation for convergence errors in random graph sequence.

| Graphon form | Operator difference | Signal difference |
|---|---|---|
| Classical form | $\Delta T_{W_n} : \|W_m - \overline{W}_{m,n}\|_{2,2}$ | $\Delta X_n : \|X_n - \overline{X}_{m,n}\|_2$ |
| Stretched form | $\Delta T_{W_n^{\mathfrak{s}}} : \|W_m^{\mathfrak{s}} - \overline{W}_{m,n}^{\mathfrak{s}}\|_{2,2}$ | $\Delta X_n^{\mathfrak{s}} : \|X_m^{\mathfrak{s}} - \overline{X}_{m,n}^{\mathfrak{s}}\|_2$ |

**Lemma 3** (Convergence to Stretched Graphon). *Consider the stretched graphon and signal ($W_m^{\mathfrak{s}}$, $X_m^{\mathfrak{s}}$), and the stretched induced graphon and signal ($\overline{W}_{m,n}^{\mathfrak{s}}$, $\overline{X}_{m,n}^{\mathfrak{s}}$) generated from them. Suppose $n$ is sufficiently large such that $\Delta T_{W_n} \leq \|W_m\|_1$, then it holds*

$$\Delta T_{W_n^{\mathfrak{s}}} \leq C_{\mathfrak{s}} \Delta T_{W_n} + C_{W,\mathfrak{s}} \Delta T_{W_n}^{1/2},$$
$$\Delta X_n^{\mathfrak{s}} \leq C_{\mathfrak{s}} \Delta X_n + C_{X,\mathfrak{s}} \Delta T_{W_n}^{1/2}, \tag{13}$$

*where $C_{\mathfrak{s}}$ depends on the stretching coefficient $s_m = \|W_m\|_1$, $C_{W,\mathfrak{s}}$ and $C_{X,\mathfrak{s}}$ both converge to constants determined by $W_m$ and $X_m$ as $n$ increases (proof in Appendix E).*

Combined with Lemma 4, which establishes an $O(n^{-1/2})$ convergence rate under the classical form, this result provides the full error bound under stretched forms. These errors arise from sampling the graph topology and node features, while stretching introduces scaling of functions and domain mismatches, captured by constants $C_{\mathfrak{s}}, C_{W,\mathfrak{s}}, C_{X,\mathfrak{s}}$. According to the double convergence in Eq.(7), these constants remain fixed while taking the limit over $n$, which is done before convergence in $t_m$, guaranteeing the overall convergence.

## 6 Experiments

To validate the transferability results on sparse graphs, we conduct experiments on the Cora dataset (Sen et al., 2008). The Cora graph contains 2708 nodes and has an edge density of 0.00144, corresponding to sparsity functions $3.9/n$ or $1.14 \log n/n$. We fix $G_{M,N}$ as the full Cora graph, and evaluate the transferability from graphs $G_{m,n}$ with arbitrary sparsity to it. Specifically, we measure the average transfer error over the full sampled graph to analyze transfer errors.

Since the sampled graphs originate from a real-world dataset, the expanding sequence $\{W_m\}$ is approximated by an expanding sequence of subgraphs $\{G_m\}$. We construct $\{G_m\}$ from Cora by adding 50 nodes at each stage in degree order. To model different sparsity levels, we define three sparsity schemes: Scheme I: $f(n) = 2/n$; Scheme II: $f(n) = 0.2 \log n/n$; Scheme III: $f(n) = \Theta(1)$ (see Fig. 4(a)). The transferability under the sparsity of Scheme I is not guaranteed in previous works (Keriven et al., 2020; 2021; Wang et al., 2023; 2024). We set the graph sizes $n$ of $\{G_{m,n}\}$ to be $\{100, 200, ..., 1600\}$. Each sampled $G_{m,n}$ is aligned with a corresponding stage $G_m$, determined by the condition $f(n) \approx e(G_m)$, where $e(G_m)$ denotes the edge density of $G_m$. Each graph $G_{m,n}$ is constructed by randomly selecting $n$ nodes and inheriting their edges and features. The edge densities of the sampled sequences are shown in Fig. 4(a).

The GCN output of the full Cora graph serves as the reference. For each sampled subgraph $G_{m,n}$, we compare the average output error across all nodes against the reference. Unlike the classical

graphon setting, which normalizes as $\Phi(\mathbf{S}_n/n, \boldsymbol{x}_n, \mathcal{H})$ (Ruiz et al., 2023), under the stretched form we adopt $\Phi(\mathbf{S}_n/\sqrt{2e_n}, \boldsymbol{x}_n, \mathcal{H})$, where $e_n$ is the number of edges in $G_{m,n}$ (Ji et al., 2024). The difference arises from the operator impacts of inducing, as shown in Lemma 5 and Lemma 8. Each subgraph is sampled 20 times to compute the mean and standard deviation. We further consider GCNs with 2 or 3 layers and 32 or 64 hidden feature dimensions.

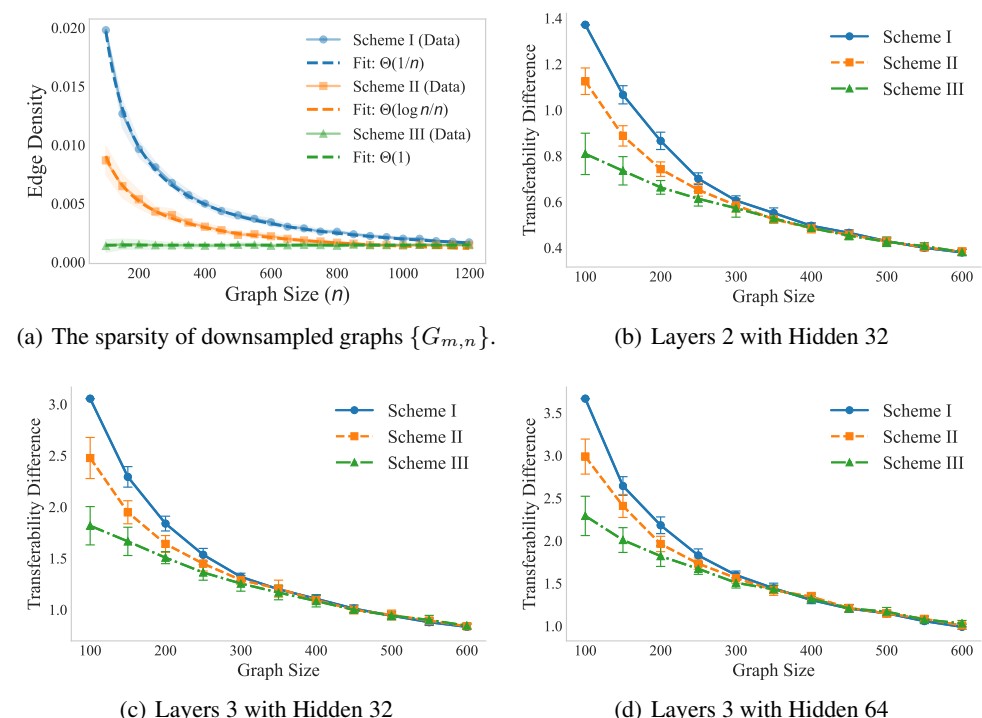

(a) The sparsity of downsampled graphs $\{G_{m,n}\}$.

(b) Layers 2 with Hidden 32

(c) Layers 3 with Hidden 32

(d) Layers 3 with Hidden 64

Figure 4: GCN transfer errors of sparse graph sequences. Transfer is performed from graphs following sparsity schemes, I: $f(n) = 2/n$; II: $f(n) = 0.2 \log n/n$; III: $f(n) = \Theta(1)$, to the target graph $G_{M,N}$, which corresponds to sparsity functions $3.9/n$ or $1.14 \log n/n$.

Across all sparsity schemes, transferability errors vanish as edge densities decrease and graph sizes increase, as shown in Figure 4. For small or medium graph sizes, sparser graphs exhibit higher edge densities, leading to larger transferability errors. For large graph sizes, sequences under different sparsity converge together, producing similar transferability errors. In Scheme III, where the expected edge density is fixed, increasing the graph size consistently reduces the transfer error. These observations validate our theoretical analysis of the roles played by edge density and graph size in transferability.

## 7 CONCLUSION

In this work, we study the transferability of GCNs across varying sparsity levels. Subgraphs are randomly generated from the original graph, resulting in multiple sparsity. By introducing SWNNs as the non-zero limit of GCNs, we establish a transfer error bound over sparse graphs, which is influenced by expected edge density and graph size, factors that jointly determine the multiple sparsity. Experiments on real-world networks validate these theoretical findings.

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

## A   THE USE OF LARGE LANGUAGE MODELS (LLMS)

We use a large language model only for language polishing. All research ideas, experiments, and analyses are conducted solely by the authors.

## B   SUPPLEMENTARY EXPERIMENTS

### B.1   PUBMED DATASET

The Pubmed graph contains 19717 nodes and has an edge density of $0.000228$, corresponding to sparsity functions $4.5/n$ or $1.05 \log n/n$. We fix $G_{M,N}$ as the full Pubmed graph, and evaluate the transferability from graphs $G_{m,n}$ to it. The average transfer error over the full sampled graph are measured to analyze transfer errors.

To model different sparsity levels, we define three sparsity schemes: Scheme I: $f(n) = 2.2/n$; Scheme II: $f(n) = 0.2 \log n/n$; Scheme III: $f(n) = \Theta(1)$ (see Fig. 5(a)). The experiments further validate our conclusion: larger graph sizes and lower edge densities lead to smaller transfer errors.

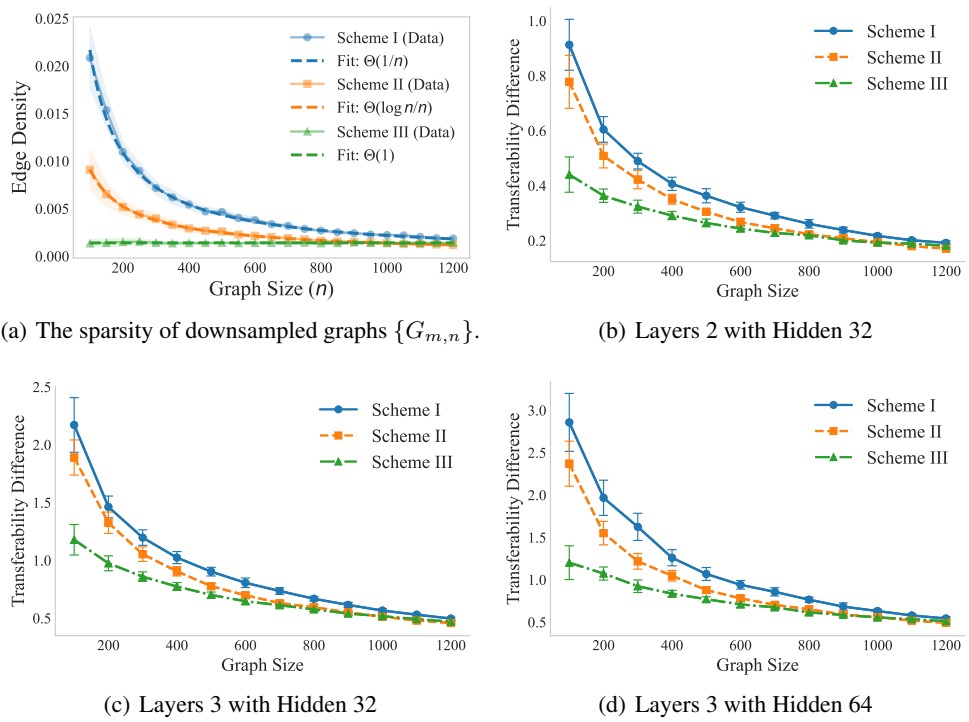

(a) The sparsity of downsampled graphs $\{G_{m,n}\}$.   (b) Layers 2 with Hidden 32

(c) Layers 3 with Hidden 32   (d) Layers 3 with Hidden 64

Figure 5: GCN transfer errors of sparse graph sequences. Transfer is performed from graphs following sparsity schemes, I: $f(n) = 2.2/n$; II: $f(n) = 0.2 \log n/n$; III: $f(n) = \Theta(1)$, to the target graph $G_{M,N}$, which corresponds to sparsity functions $4.5/n$ or $1.05 \log n/n$.

### B.2   OGBN-ARXIV DATASET

The Ogbn-Arxiv graph contains 169,343 nodes and has an edge density of $0.000081$, corresponding to sparsity functions $13.7/n$ or $1.14 \log n/n$. We fix $G_{M,N}$ as the full Ogbn-Arxiv graph, and evaluate the transferability from graphs $G_{m,n}$ to it. The average transfer error over the full sampled graph are measured to analyze transfer errors.

To model different sparsity levels, we define three sparsity schemes: Scheme I: $f(n) = 8/n$; Scheme II: $f(n) = \log n/n$; Scheme III: $f(n) = \Theta(1)$ (see Fig. 6(a)). The experiments further validate our conclusion: larger graph sizes and lower edge densities lead to smaller transfer errors.

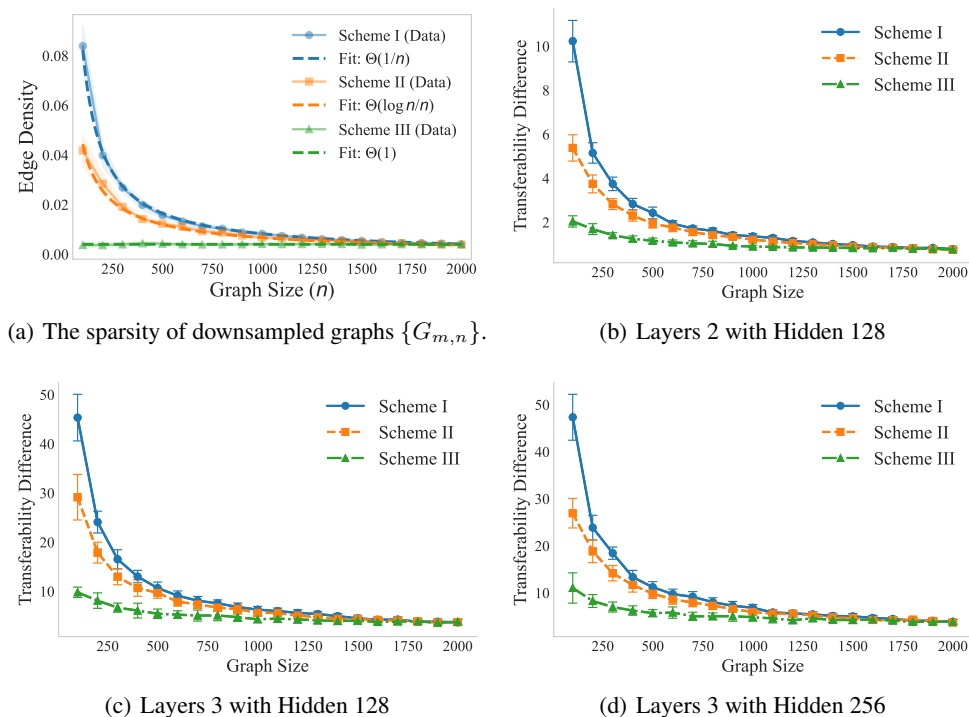

(a) The sparsity of downsampled graphs $\{G_{m,n}\}$.

(b) Layers 2 with Hidden 128

(c) Layers 3 with Hidden 128

(d) Layers 3 with Hidden 256

Figure 6: GCN transfer errors of sparse graph sequences. Transfer is performed from graphs following sparsity schemes, I: $f(n) = 10/n$; II: $f(n) = \log n/n$; III: $f(n) = \Theta(1)$, to the target graph $G_{M,N}$, which corresponds to sparsity functions $13.7/n$ or $1.14 \log n/n$.

## C    APPENDIX: GENERALIZED CONCEPTS FOR ANALYSIS

In some graphon analyses, new functions are constructed, such as $\mathcal{W} = W_0 - W_1$, the value range of which exceeds the probability range $[0, 1]$ in Definition 1. To make our proof more concise and clear, we introduce several related generalized concepts.

**Definition 3** (Value-unbounded Graphon). *By just extending the value range of graphon $[0, 1]$ into $\mathbb{R}$, the value-unbounded graphon is denoted by $\mathcal{W} : \mathbb{R}_+^2 \to \mathbb{R}$.*

Similarly, we consider not only the adjacency matrix $\mathbf{S}_n \in \{0, 1\}^{n \times n}$ consisting of binary entries, but also symmetric matrices with wider range of values, denoted by $\mathcal{S}_n \in \mathbb{R}^{n \times n}$, whose induced functional form is given by

$$\overline{\mathcal{W}}_n(u \in \mathbb{I}_i, v \in \mathbb{I}_j) = \mathcal{S}_n(i, j), \tag{14}$$

which is only supported over $[0, 1]^2$.

In Definition 2, stretching is defined using a specific coefficient $s = \sqrt{|W|_1}$, which is associated with graphon $W$ to be stretched. Accordingly, we introduce a more general form of stretching, in which the coefficient is not restricted to any specific graphon, denoted by $s_\forall \in (0, 1]$.

**Definition 4** (Non-specific Stretching). *For value-unbounded graphon $\mathcal{W}(u, v)$, the stretched graphon $\mathcal{W}^{s_\forall}$ and the stretched graphon signal is defined by,*

$$\begin{aligned} \mathcal{W}^{s_\forall}(u, v) &= \mathcal{W}\left(s_\forall u, s_\forall v\right), \\ X^{s_\forall}(u) &= X\left(s_\forall u\right), \end{aligned} \tag{15}$$

*where $s_\forall \in (0, 1]$ is non-specific stretching coefficient.*

The non-specific stretching of classical graphon and signal is similar to Eq.( 6), by replacing the specific $s$ with non-specific $s_\forall$.

## D  Proof of Lemma 2: Sparse Graphon Sequence

We write down the constants or values approaching constants generated in the following proofs of this section as follows, where we require $k$ to satisfy $k' = (k-3)/2 > 0$:

$$C_{k,m} = \frac{2C_k}{(k-1)^2},$$

$$C_{k',W} = \frac{C_{k,m}}{s_R} + A_W \sqrt{\frac{21}{8}} \frac{C_{k,m}}{s_R^3} + \frac{\sqrt{2C_{k,m}}}{s_R^2}, \tag{16}$$

$$C_{k',X} = A_X \frac{2}{\sqrt{27}} \frac{C_{k,m}}{s_R^2} + \sqrt{\frac{4\max(X)C_{k,m}}{3s_R^3}}.$$

The equivalence between $W_m^{\mathfrak{s}}$ and $W_m'^{\mathfrak{s}}$, and the equivalence between $X_m^{\mathfrak{s}}$ and $X_m'^{\mathfrak{s}}$ have been described in Eq.( 17).

$$W_m^{\mathfrak{s}}(u,v) = W_m(\frac{s_m'}{t_m}u, \frac{s_m'}{t_m}v) = W_m'(s_m'u, s_m'v) = W_m'^{\mathfrak{s}}(u,v),$$

$$X_m^{\mathfrak{s}}(u) = X_m(\frac{s_m'}{t_m}u) = X_m'(s_m'u) = X_m'^{\mathfrak{s}}(u). \tag{17}$$

Therefore, the convergence of $W_m^{\mathfrak{s}}$ and $X_m^{\mathfrak{s}}$ is equivalent to that of $W_m'^{\mathfrak{s}}$ and $X_m'^{\mathfrak{s}}$. Therefore, in the following discussion, we focus directly on the convergence of $W_m'^{\mathfrak{s}}$ and $X_m'^{\mathfrak{s}}$.

### D.1  Stretching Coefficients of Sparse Graphon

We denote the stretching coefficients of $W_m'^{\mathfrak{s}}$ and $W_{\mathbb{R}_+}^{\mathfrak{s}}$ by $s_m'$ and $s_R$. According to the definition of stretching and the truncation process, we have

$$\left| s_R^2 - s_m'^2 \right| = \left| \|W_{\mathbb{R}_+}\|_1 - \|W_m'\|_1 \right| = \|W_{\mathbb{R}_+}\|_{L^1(\mathbb{R}_+^2/[0,t_m]^2)}. \tag{18}$$

Considering that $W_{\mathbb{R}_+}$ belongs to the Schwartz space, i.e., for any $k > 0$, there exits a constant $C_k$ such that

$$W_{\mathbb{R}_+}(u,v) \le \frac{C_k}{(u+1)^k(v+1)^k}. \tag{19}$$

Based on it, we derive the upper bound of $\left| s_R^2 - s_m'^2 \right|$:

$$\left| s_R^2 - s_m'^2 \right| \le \left\| \frac{C_k}{(u+1)^k(v+1)^k} \right\|_{L^1(\mathbb{R}_+^2/[0,t_m]^2)}$$

$$\le \int_{u \ge t_m} \int_{\mathbb{R}_+} \frac{C_k}{(u+1)^k(v+1)^k} dv du + \int_{\mathbb{R}_+} \int_{v \ge t_m} \frac{C_k}{(u+1)^k(v+1)^k} dv du \tag{20}$$

$$= 2 \int_{u \ge t_m} \int_{\mathbb{R}_+} \frac{C_k}{(u+1)^k(v+1)^k} dv du.$$

By setting $k > 1$, the integral in the above expression can be evaluated, yielding

$$\left| s_R^2 - s_m'^2 \right| \le \frac{2C_k}{(k-1)^2}(1+t_m)^{-k+1} \le \frac{2C_k}{(k-1)^2} \frac{1}{t_m^{k-1}}. \tag{21}$$

### D.2  Stretched Graphon of Sparse Graphon

For clarity, we denote the stretched forms of $W_{\mathbb{R}_+}$ and $W_m'$ by $W_{\mathbb{R}_+}^{\mathfrak{s}_R}$ and $W_m'^{\mathfrak{s}_m'}$, so that the stretch defined with specific coefficients can be distinguished from that defined with non-specific coefficients. In the proof, we introduce an intermediate term $W_m'^{\mathfrak{s}_R}$ according to Definition 4. By the triangle inequality, we have:

$$\left\| W_{\mathbb{R}_+}^{\mathfrak{s}} - W_m^{\mathfrak{s}} \right\|_{2,2} \le \left\| W_{\mathbb{R}_+}^{\mathfrak{s}_R} - W_m'^{\mathfrak{s}_R} \right\|_{2,2} + \left\| W_m'^{\mathfrak{s}_R} - W_m'^{\mathfrak{s}_m'} \right\|_{2,2}$$

$$\le \left\| W_{\mathbb{R}_+}^{\mathfrak{s}_R} - W_m'^{\mathfrak{s}_R} \right\|_2 + \left\| W_m'^{\mathfrak{s}_R} - W_m'^{\mathfrak{s}_m'} \right\|_2, \tag{22}$$

the transition from the operator norm to the $L^2$-norm relies on Lemma 1 in Ji et al. (2024).

For the first part, since $W'_m$ is directly obtained from $W_{\mathbb{R}_+}$ by truncation, and both share the same stretching factor according to Definition 4, we have

$$W'^{s_R}_m(u, v) = W^{s_R}_{\mathbb{R}_+}(u, v), \quad (u, v) \in [0, t_m/s_R]^2.$$

Hence, noting that $W_{\mathbb{R}_+} \leq 1$, this part reduces to

$$
\begin{aligned}
\left\| W^{s_R}_{\mathbb{R}_+} - W'^{s_R}_m \right\|_2 &= \frac{1}{s_R} \left\| W_{\mathbb{R}_+} \right\|_{L^2(\mathbb{R}^2_+/[0,t_m]^2)} \\
&\leq \frac{1}{s_R} \left\| W_{\mathbb{R}_+} \right\|_{L^1(\mathbb{R}^2_+/[0,t_m]^2)} = \frac{1}{s_R} \left| s_R^2 - s'^2_m \right|.
\end{aligned}
\tag{23}
$$

For the second part, it should be noted that $W'^{s_R}_m$ and $W'^{s'_m}_m$ are supported on different domains, namely $[0, t_m/s_R]^2$ and $[0, t_m/s'_m]^2$, respectively. Consequently, when evaluating their discrepancy in terms of the $L^2$-norm, the integration has to be considered separately over the corresponding regions:

$$
\begin{aligned}
\left\| W'^{s_R}_m - W'^{s'_m}_m \right\|^2_2 &= \int \int_{\mathbb{R}^2_+} \left( W'^{s_R}_m(u, v) - W'^{s'_m}_m(u, v) \right)^2 du\,dv \\
&= \underbrace{\left\| W'^{s_R}_m - W'^{s'_m}_m \right\|^2_{L^2([0,t_m/s_R]^2)}}_{\text{(i)}} + \underbrace{\left\| W'^{s'_m}_m \right\|^2_{L^2([0,t_m/s'_m]^2/[0,t_m/s_R]^2)}}_{\text{(ii)}},
\end{aligned}
\tag{24}
$$

where the second term accounts for the mismatch of domains.

By expanding the integral of the first term in the above expression in detail, we obtain

$$
\begin{aligned}
\underbrace{\left\| W'^{s_R}_m - W'^{s'_m}_m \right\|^2_{L^2([0,t_m/s_R]^2)}}_{\text{(i)}} &= \int \int_{[0,\frac{t_m}{s_R}]^2} \left( W_{\mathbb{R}_+}(s_R u, s_R v) - W_{\mathbb{R}_+}(s'_m u, s'_m v) \right)^2 du\,dv \\
&\leq A^2_W \int \int_{[0,\frac{t_m}{s_R}]^2} \left( |s_R u - s'_m u| + |s_R v - s'_m v| \right)^2 du\,dv \\
&= A^2_W \frac{7 t^4_m}{6 s^4_R} (s_R - s'_m)^2,
\end{aligned}
\tag{25}
$$

where the inequality is derived by the Lipschitz property of $W_{\mathbb{R}_+}$.

For the second term (ii), since the maximum value of the function is 1, the integral result is obtained by evaluating the mismatch area,

$$
\underbrace{\left\| W'^{s'_m}_m \right\|^2_{L^2([0,t_m/s'_m]^2/[0,t_m/s_R]^2)}}_{\text{(ii)}} \leq \left( \frac{t_m}{s'_m} \right)^2 - \left( \frac{t_m}{s_R} \right)^2 = \frac{t^2_m}{s'^2_m s^2_R} \left( s^2_R - s'^2_m \right).
\tag{26}
$$

Assume $t_m$ is sufficiently large such that $\|W'_m\|_1 \geq \|W_{\mathbb{R}_+}\|_1/2$, then the above inequalities becomes

$$
\begin{aligned}
\underbrace{\left\| W'^{s_R}_m - W'^{s'_m}_m \right\|^2_{L^2([0,t_m/s_R]^2)}}_{\text{(i)}} &\leq A^2_W \frac{63 t^4_m}{24 s^6_R} (s^2_R - s'^2_m)^2, \\
\underbrace{\left\| W'^{s'_m}_m \right\|^2_{L^2([0,t_m/s'_m]^2/[0,t_m/s_R]^2)}}_{\text{(ii)}} &\leq \frac{2 t^2_m}{s^4_R} \left( s^2_R - s'^2_m \right).
\end{aligned}
\tag{27}
$$

Substituting inequality 21 into the above expression 27, in order to offset the effect of $t_m$ in the numerator, we impose the condition $k - 1 > 2$. By further combining this with the previous in-

equality 23, we obtain

$$
\left\| W_{\mathbb{R}_+}^{\mathfrak{s}} - W_m^{\mathfrak{s}} \right\|_{2,2} \leq \frac{1}{s_R} \left| s_R^2 - s_m'^2 \right| + A_W \sqrt{\frac{63}{24 s_R^6}} t_m^2 \left| s_R^2 - s_m'^2 \right| + \sqrt{\frac{2}{s_R^4}} t_m \sqrt{|s_R^2 - s_m'^2|}
$$

$$
\leq \frac{C_{k,m}}{s_R} \frac{1}{t_m^{k-1}} + A_W \sqrt{\frac{63}{24 s_R^6}} \frac{C_{k,m}}{t_m^{k-3}} + \sqrt{\frac{2}{s_R^4}} \frac{\sqrt{C_{k,m}}}{t_m^{(k-3)/2}}
\tag{28}
$$

where $C_{k,m} = \frac{2 C_k}{(k-1)^2}$ denotes a general form of the constant in the inequality $\left| s_R^2 - s_m'^2 \right| \leq \frac{2 C_k}{(k-1)^2} \frac{1}{t_m^{k-1}}$ $(k > 1)$. To simplify the presentation, we replace $(k-3)/2$ with $k' = (k-3)/2$, and by applying contraction $t_m^{-(k-1)} \leq t_m^{-(k-3)} \leq t_m^{-(k-3)/2}$, the above result is transformed into

$$
\left\| W_{\mathbb{R}_+}^{\mathfrak{s}} - W_m^{\mathfrak{s}} \right\|_{2,2} \leq \frac{C_{k',W}}{t_m^{k'}},
\tag{29}
$$

where $k' > 0$ and $C_{k',W} = \frac{C_{k,m}}{s_R} + A_W \sqrt{\frac{63}{24 s_R^6}} C_{k,m} + \frac{\sqrt{2 C_{k,m}}}{s_R^2}$.

### D.3 Stretched Signal of Sparse Graphon

For clarify, We denote the stretched forms of $X_{\mathbb{R}_+}$ and $X_m'$ by $X_{\mathbb{R}_+}^{\mathfrak{s}_R}$ and $X_m'^{\mathfrak{s}_m'}$. Based on Definition 4, we introduce an intermediate term $X_m'^{\mathfrak{s}_R}$. By the triangle inequality, we have:

$$
\left\| X_{\mathbb{R}_+}^{\mathfrak{s}} - X_m^{\mathfrak{s}} \right\|_2 \leq \left\| X_{\mathbb{R}_+}^{\mathfrak{s}_R} - X_m'^{\mathfrak{s}_R} \right\|_2 + \left\| X_m'^{\mathfrak{s}_R} - X_m'^{\mathfrak{s}_m'} \right\|_2
$$

$$
\leq \left\| X_{\mathbb{R}_+}^{\mathfrak{s}_R} - X_m'^{\mathfrak{s}_R} \right\|_2 + \left\| X_m'^{\mathfrak{s}_R} - X_m'^{\mathfrak{s}_m'} \right\|_2.
\tag{30}
$$

For the first part, since $X_m'$ is directly obtained from $X_{\mathbb{R}_+}$ by truncation, and both share the same stretching coefficient, we have

$$
X_m'^{\mathfrak{s}_R}(u) = X_{\mathbb{R}_+}^{\mathfrak{s}_R}(u), \quad u \in [0, t_m/s_R].
$$

Hence, this part reduces to

$$
\left\| X_{\mathbb{R}_+}^{\mathfrak{s}_R} - X_m'^{\mathfrak{s}_R} \right\|_2 = \frac{1}{\sqrt{s_R}} \left\| X_{\mathbb{R}_+} \right\|_{L^2(\mathbb{R}_+ / [0, t_m])}.
\tag{31}
$$

For the second part, $X_m'^{\mathfrak{s}_R}$ and $X_m'^{\mathfrak{s}_m'}$ are supported on $[0, t_m/s_R]$ and $[0, t_m/s_m']$ differently. Consequently, it's considered separately over the corresponding regions:

$$
\left\| X_m'^{\mathfrak{s}_R} - X_m'^{\mathfrak{s}_m'} \right\|_2^2 = \int_{\mathbb{R}_+} \left( X_m'^{\mathfrak{s}_R}(u) - X_m'^{\mathfrak{s}_m'}(u) \right)^2 du
$$

$$
= \underbrace{\left\| X_m'^{\mathfrak{s}_R} - X_m'^{\mathfrak{s}_m'} \right\|_{L^2([0, t_m/s_R])}^2}_{\textbf{(i)}} + \underbrace{\left\| X_m'^{\mathfrak{s}_m'} \right\|_{L^2([t_m/s_m', t_m/s_R])}^2}_{\textbf{(ii)}},
\tag{32}
$$

where the second term accounts for the mismatch of domains. Expanding the integral of the first term in detail, we obtain

$$
\underbrace{\left\| X_m'^{\mathfrak{s}_R} - X_m'^{\mathfrak{s}_m'} \right\|_{L^2([0, t_m/s_R])}^2}_{\textbf{(i)}} = \int_{[0, \frac{t_m}{s_R}]} \left( X_{\mathbb{R}_+}(s_R u) - X_{\mathbb{R}_+}(s_m' u) \right)^2 du
$$

$$
\leq A_X^2 \int_{[0, \frac{t_m}{s_R}]} (s_R u - s_m' u)^2 du
\tag{33}
$$

$$
= A_X^2 \frac{t_m^2}{3 s_R^2} (s_R - s_m')^2,
$$

where the inequality is derived by the Lipschitz property of $X_{\mathbb{R}_+}$. For the second term (ii), we denote the maximum value of $X_{\mathbb{R}_+}$ by $\max(X_{\mathbb{R}_+})$, then the integral result is obtained by evaluating the mismatch area,

$$\underbrace{\left\| X_m'^{s_m'} \right\|_{L^2([t_m/s_m', t_m/s_R])}^2}_{\text{(ii)}} \leq \max(X) \left( \frac{t_m}{s_R} - \frac{t_m}{s_m'} \right) = \frac{\max(X)t_m}{s_m' s_R}(s_R - s_m'). \qquad (34)$$

When $t_m$ is sufficiently large such that $s_m'^2 \geq s_R^2/2$, then the above inequalities becomes

$$
\begin{aligned}
&\underbrace{\left\| X_m'^{s_R} - X_m'^{s_m'} \right\|_{L^2([0, t_m/s_R])}^2}_{\text{(i)}} \leq A_X^2 \frac{4t_m^2}{27 s_R^4}(s_R^2 - s_m'^2)^2, \\
&\underbrace{\left\| X_m'^{s_m'} \right\|_{L^2([0, t_m/s_m']/[0, t_m/s_R])}^2}_{\text{(ii)}} \leq \frac{4\max(X)t_m}{3 s_R^3}\left( s_R^2 - s_m'^2 \right).
\end{aligned}
\qquad (35)
$$

Similarly, to offset the effect of $t_m$ in the numerator, we impose the condition $k - 1 > 1$ of the inequality 21. Combining this with the previous results, we obtain

$$
\begin{aligned}
\left\| X_{\mathbb{R}_+}^{\mathfrak{s}} - X_m^{\mathfrak{s}} \right\|_2 &\leq \frac{1}{\sqrt{s_R}} \left\| X_{\mathbb{R}_+} \right\|_{L^2(\mathbb{R}_+/[0,t_m])} \\
&\quad + A_X \sqrt{\frac{4}{27 s_R^4}} t_m \left| s_R^2 - s_m'^2 \right| + \sqrt{\frac{4\max(X)}{3 s_R^3}} \sqrt{t_m \left| s_R^2 - s_m'^2 \right|} \\
&\leq \frac{\left\| X_{\mathbb{R}_+} \right\|_{L^2([t_m, +\infty))}}{\sqrt{s_R}} + A_X \sqrt{\frac{4}{27 s_R^4}} \frac{C_{k,m}}{t_m^{k-2}} + \sqrt{\frac{4\max(X)}{3 s_R^3}} \frac{\sqrt{C_{k,m}}}{t_m^{(k-2)/2}}.
\end{aligned}
\qquad (36)
$$

To simplify the presentation, we replace $(k-2)/2$ with $k' = (k-2)/2$, and by applying contraction $t_m^{-(k-2)} \leq t_m^{-(k-2)/2}$, the above result is transformed into

$$\left\| X_{\mathbb{R}_+}^{\mathfrak{s}} - X_m^{\mathfrak{s}} \right\|_2 \leq \frac{C_{k',X}}{t_m^{k'}} + \frac{\left\| X_{\mathbb{R}_+} \right\|_{L^2([t_m, +\infty))}}{\sqrt{s_R}}, \qquad (37)$$

where $k' > 0$ and $C_{k',X} = A_X \sqrt{\frac{4}{27 s_R^4}} C_{k,m} + \sqrt{\frac{4\max(X)C_{k,m}}{3 s_R^3}}$.

## E  PROOF OF LEMMA 3 AND LEMMA 4: RANDOM GRAPH SEQUENCE

**To simplify notation**, throughout this part of the proof we write $W, \overline{W}_n$ instead of the full symbols $W_m, \overline{W}_{m,n}$, and similarly $X, \overline{X}_n$ for $X_m, \overline{X}_{m,n}$. The rationale is that when a graph sequence converges to the graphon and graphon signal, the graphon and signal is considered invariant.

We also write down the constants or values approaching constants generated in the following proofs of this section as follows, where $n$ is sufficiently large such that $\Delta T_{W_n} \leq s_m^2/2$:

$$
\begin{aligned}
&C_W = \frac{C_\nu}{t_m} + \frac{2A_W}{\sqrt{6}\epsilon_u}, \quad C_X = \frac{2A_X}{\sqrt{6}\epsilon_u}. \\
&C_{W,1} = \frac{2\sqrt{7}A_W t_m}{\sqrt{3}s_m^3(1 + 1/\sqrt{2})}, \quad C_{W,2} = \frac{2}{s_m^2\sqrt{(1 + 1/\sqrt{2})/\sqrt{2}}}. \\
&C_{X,1} = \frac{2^{5/4}A_X t_m}{\sqrt{3}s_m^{5/2}(1 + 1/\sqrt{2})}, \quad C_{X,2} = \frac{2^{5/8}\max(|X|)}{\sqrt{s_m^{5/2}/\sqrt{2}(1 + 1/\sqrt{2})}} \\
&C_{\mathfrak{s}} := \max\left\{ \frac{\sqrt{2}}{s_m}, \frac{2^{\frac{1}{4}}}{\sqrt{s_m}} \right\}, \\
&C_{W,\mathfrak{s}} = C_{W,1}\Delta T_{W_n}^{1/2} + C_{W,2} \rightarrow C_{W,2}, \quad C_{X,\mathfrak{s}} = C_{X,1}\Delta T_{W_n}^{1/2} + C_{X,2} \rightarrow C_{X,2}.
\end{aligned}
\qquad (38)
$$

$C_{\mathfrak{s}} := \max\{\frac{\sqrt{2}}{s}, \frac{2^{\frac{1}{4}}}{\sqrt{s}}\}$ is constant about $W$, $C_{W,\mathfrak{s}} = C_{W,1}\Delta T_{W_n}^{1/2} + C_{W,2}$ and $C_{X,\mathfrak{s}} = C_{X,1}\Delta T_{W_n}^{1/2} + C_{X,2}$ converge to constants $C_{W,2}$ and $C_{X,2}$ as $n$ increases.

### E.1 PROOF OF LEMMA 4: CLASSICAL FORM

**Lemma 4.** *(Convergence to Classical Graphon)*

*Consider the classical graphon and signal $(W_m, X_m)$, and the induced graphon and signal $(\overline{W}_{m,n}, \overline{X}_{m,n})$ sampled from them. For any $\nu > 0$, assume $n$ is large enough such that $n^{-\nu} \le \epsilon_b$, there exists a constant $C_\nu$ such that*

$$\Delta T_{W_n} \le \frac{C_W}{\sqrt{n}} t_m, \ \Delta X_n \le \frac{C_X}{\sqrt{n}} t_m \tag{39}$$

*with probability at least $(1 - \epsilon_u)(1 - \epsilon_b)$, where $\epsilon_u, \epsilon_b \in (0, 1)$ are related to the Uniform sampling and Bernoulli sampling, $C_W = \frac{C_\nu}{t_m} + \frac{2A_W}{\sqrt{6}\epsilon_u}$ and $C_X = \frac{2A_X}{\sqrt{6}\epsilon_u}$ are constants about $W_m$ and $X_m$.*

Under the classical graphon form, as the graph size increases, the random graphs converge to the limit $W_m, X_m$ at the rate of $O(1/n^{1/2})$. However, when $W_m$ diminishes due to sparsity, the full random graph sequence converges to the trivial zero limit.

For the graph topological sampling process, modeled as $W \sim \overline{P}_n \sim \overline{W}_n$, where $\overline{P}_n$ is the induced graphon of the probability matrix. The procedure involves two steps: first, a probability matrix $\mathbf{P}_n$ is sampled from $W$; second, an adjacency matrix $\mathbf{S}_n$ is sampled from the probability matrix. In the graph signal sampling process, where $X \sim \overline{X}_n$, the discrepancy arises due to uniform sampling. Accordingly, the difference can be decomposed into two components:

- Uniform Sampling: $\left\|X - \overline{X}_n\right\|_2$, $\left\|W - \overline{P}_n\right\|_{2,2}$.

- Bernoulli Sampling: $\left\|\overline{P}_n - \overline{W}_n\right\|_{2,2}$.

As the graph size $n$ increases, the graph and graph signal converge to the corresponding graphon and graphon signal. As we do not consider the effect of different vertex permutations of the same graph, we assume that the vertices are ordered according to their latent features, such that $u_1 \le u_2 \le ... \le u_n$. To provide a clear exposition, we decompose the conclusion of Lemma 4 into two components: convergence of graph signals E.1.2 and convergence of graph structures E.1.3.

Notably, $W_{\mathbb{R}_+}$ and $X_{\mathbb{R}_+}$ satisfy the $A_W$-Lipschitz and $A_X$-Lipschitz properties. In contrast, $W_m$ and $X_m$, being derived through stretching, satisfy the corresponding $t_m A_W$-Lipschitz and $t_m A_X$-Lipschitz properties.

#### E.1.1 NORMALIZATION OF CLASSICAL INDUCING

**Lemma 5.** *Let $(\overline{\mathcal{W}}_n, \overline{X}_n)$ be the induced form of value-unbounded graph matrix $\mathcal{S}_n$ $(\mathcal{S}_n \in \mathbb{R}^{n \times n})$ and node features $\boldsymbol{x}_n$. Then it holds*

$$\left\|\overline{X}_n\right\|_2 = \frac{1}{\sqrt{n}}\|\boldsymbol{x}_n\|_2,$$

$$\left\|\overline{\mathcal{W}}_n\right\|_{2,2} = \frac{1}{n}\|\mathcal{S}_n\|_{2,2}. \tag{40}$$

*Proof.* For the $L^2$-norm property between $\overline{X}_n$ and $\boldsymbol{x}_n$,

$$\left\|\overline{X}_n\right\|_2^2 = \int_{[0,1]} \overline{X}_n^2(u)du = \sum_{i=1}^{n} \frac{1}{n}\overline{X}_n^2(u \in \mathbb{I}_i)$$

$$= \frac{1}{n}\sum_{i=1}^{n} \boldsymbol{x}_n^2(i) = \frac{1}{n}\|\boldsymbol{x}_n\|_2^2. \tag{41}$$

For the operator norm property, according to the definition of $\left\|\overline{\mathcal{W}}_n\right\|_{2,2}$, we need to calculate the supremum of $\frac{\left\|T_{\overline{\mathcal{W}}_n}X\right\|_2}{\|X\|_2}$ for all $X \in L^2([0,1])$.

$$\frac{\left\|T_{\overline{\mathcal{W}}_n}X\right\|_2^2}{\|X\|_2^2} = \frac{\int_{[0,1]}\left(\int_{[0,1]}\overline{\mathcal{W}}_n(u,v)X(u)du\right)^2 dv}{\|X\|_2^2}$$
$$= \frac{\int_{[0,1]}\left(\sum_{i=1}^n \overline{\mathcal{W}}_n(u \in \mathbb{I}_i, v)\int_{\mathbb{I}_i} X(u)du\right)^2 dv}{\|X\|_2^2}. \tag{42}$$

We construct $\overline{X}_n(u \in \mathbb{I}_i) = n\int_{\mathbb{I}_i} X(u)du$, taking the mean of $X$ over each partition $\mathbb{I}_i$, where the length is $1/n$. Then the above equality becomes

$$\frac{\left\|T_{\overline{\mathcal{W}}_n}X\right\|_2^2}{\|X\|_2^2} = \frac{\int_{[0,1]}\left(\sum_{i=1}^n \overline{\mathcal{W}}_n(u \in \mathbb{I}_i, v)\overline{X}_n(u \in \mathbb{I}_i)/n\right)^2 dv}{\|X\|_2^2}$$
$$= \frac{\left\|T_{\overline{\mathcal{W}}_n}\overline{X}_n\right\|_2^2}{\|X\|_2^2}. \tag{43}$$

Comparing $\left\|\overline{X}_n\right\|_2^2$ and $\|X\|_2^2$, we have

$$\|X\|_2^2 = \int_{[0,1]} X^2(u)du = \sum_{i=1}^n \int_{\mathbb{I}_i} X^2(u)du$$
$$\geq \sum_{i=1}^n n\left(\int_{\mathbb{I}_i} X(u)1(u)du\right)^2$$
$$= \sum_{i=1}^n \frac{1}{n}\left(n\int_{\mathbb{I}_i} X(u)du\right)^2$$
$$= \left\|\overline{X}_n\right\|_2^2, \tag{44}$$

which is derived by the Cauchy-Schwarz inequality with $\int_{\mathbb{I}_i} 1^2(u)du = 1/n$. Therefore, for any $X \in L^2([0,1])$, it holds

$$\frac{\left\|T_{\overline{\mathcal{W}}_n}X\right\|_2^2}{\|X\|_2^2} \leq \frac{\left\|T_{\overline{\mathcal{W}}_n}\overline{X}_n\right\|_2^2}{\left\|\overline{X}_n\right\|_2^2}. \tag{45}$$

For each $X \in L^2([0,1])$, we can derive a $\overline{X}_n \in L^2([0,1])$ from $X$ to make the above inequality hold. Then the operator norm becomes

$$\left\|\overline{\mathcal{W}}_n\right\|_{2,2}^2 := \sup_{\forall \overline{X}_n \in L^2([0,1])} \frac{\left\|T_{\overline{\mathcal{W}}_n}\overline{X}_n\right\|_2^2}{\left\|\overline{X}_n\right\|_2^2}, \tag{46}$$

although $\overline{X}_n$ is derived from $X$, obviously $\overline{X}_n$ can take all possible values within its defined space. Therefore, $\overline{X}_n$ can correspond to every $\boldsymbol{x}_n \in \mathbb{R}^n$.

$$\frac{\left\|T_{\overline{\mathcal{W}}_n}\overline{X}_n\right\|_2^2}{\left\|\overline{X}_n\right\|_2^2} = \frac{\int_{[0,1]}\left(\sum_{i=1}^n \overline{\mathcal{W}}_n(u \in \mathbb{I}_i, v)\overline{X}_n(u \in \mathbb{I}_i)/n\right)^2 dv}{\left\|\overline{X}\right\|_2^2}$$
$$= \frac{\sum_{j=1}^n \left(\sum_{i=1}^n \mathcal{S}_n(i,j)\boldsymbol{x}_n(i)/n\right)^2/n}{\|\boldsymbol{x}\|_2^2/n} \tag{47}$$
$$= \frac{1}{n^2}\frac{\left\|\mathcal{S}_n\boldsymbol{x}_n\right\|_2^2}{\|\boldsymbol{x}_n\|_2^2}.$$

As all $\boldsymbol{x}_n \in \mathbb{R}^n$ are considered in the above equality, we have

$$\left\|\overline{\mathcal{W}}_n\right\|_{2,2}^2 := \sup_{\forall \boldsymbol{x}_n \in \mathbb{R}^n} \frac{1}{n^2}\frac{\left\|\mathcal{S}_n\boldsymbol{x}_n\right\|_2^2}{\|\boldsymbol{x}_n\|_2^2} = \frac{1}{n^2}\left\|\mathcal{S}_n\right\|_{2,2}^2. \tag{48}$$

$\square$

### E.1.2 CLASSICAL SIGNAL CONVERGENCE

**Lemma.** *(Convergence to Graphon Signal)*

*Consider the graphon signal $X_m$ generated from $X_{\mathbb{R}_+}$ that satisfies AS2, and the induced graphon signal $\overline{X}_{m,n}$ sampled from $X_m$, then the following holds*

$$\Delta X_n \leq \frac{A_X t_m}{\sqrt{6n}\epsilon_u} \tag{49}$$

*with probability at least $1 - \epsilon_u$.*

*Proof.* The induced graphon signal $\overline{X}_n$ is defined on $n$ equal partitions of $[0, 1]$, denoted by $\{\mathbb{I}_i | 1 \leq i \leq n\}$, i.e. $\mathbb{I}_i = [\frac{i-1}{n}, \frac{i}{n})$ for $1 \leq i \leq n-1$, $\mathbb{I}_n = [\frac{n-1}{n}, 1]$. Therefore, $\|\overline{X}_n - X\|_2$ can be divided into $n$ parts,

$$\left\|\overline{X}_n - X\right\|_2^2 = \sum_{i=1}^n \int_{\mathbb{I}_i} (X(u_i) - X(u))^2 du. \tag{50}$$

By using Jensen's inequality and the $t_m A_X$-Lipschitz property, we have

$$\begin{aligned}
\left\{\mathbb{E}\left[\|\overline{X}_n - X\|_2\right]\right\}^2 &\leq \mathbb{E}\left[\|\overline{X}_n - X\|_2^2\right] \\
&= \mathbb{E}\left[\sum_{i=1}^n \int_{\mathbb{I}_i} (X(u_i) - X(u))^2 du\right] \\
&\leq A_X^2 t_m^2 \mathbb{E}\left[\sum_{i=1}^n \int_{\mathbb{I}_i} (u_i - u)^2 du\right].
\end{aligned} \tag{51}$$

For the expectation term of the above right side, calculate the integral of $u$,

$$\begin{aligned}
\mathbb{E}\left[\sum_{i=1}^n \int_{\mathbb{I}_i} (u_i - u)^2 du\right] &= \mathbb{E}\left\{\sum_{i=1}^n \frac{i^3 - (i-1)^3}{3n^3} - \frac{2i-1}{n^2} u_i + \frac{1}{n} u_i^2\right\} \\
&= \frac{1}{3} - \frac{1}{n^2}\sum_{i=1}^n (2i-1)\mathbb{E}[u_i] + \frac{1}{n}\sum_{i=1}^n \mathbb{E}[u_i^2].
\end{aligned} \tag{52}$$

According to the order statistic theory over $n$ i.i.d. uniform distributions, the probability density of $u_i$ $(1 \leq i \leq n)$ is $\text{Beta}(i, n - i + 1)$ Okoyo (2016), that is

$$\mathbb{E}[u_i] = \frac{i}{n+1}, \quad \mathbb{E}[u_i^2] = \frac{i^2 + i}{(n+1)(n+2)}, \tag{53}$$

subtitute the above equations into (52), and calculate the sum,

$$\mathbb{E}\left[\sum_{i=1}^n \int_{\mathbb{I}_i} (u_i - u)^2 du\right] = \frac{1}{6n}. \tag{54}$$

Combining (51) and (54), it becomes $\mathbb{E}\left[\|\overline{X}_n - X\|_2\right] \leq \frac{A_X t_m}{\sqrt{6n}}$. Using the Markov inequality, we have

$$P\left(\|\overline{X}_n - X\|_2 \geq \frac{A_X t_m}{\sqrt{6n}\epsilon_u}\right) \leq \epsilon_u. \tag{55}$$

$\square$

### E.1.3 CLASSICAL GRAPHON CONVERGENCE

**Lemma 6.** *Lei & Rinaldo (2015) Considering the probability matrix $\mathbf{P}_n$ and the adjacency matrix $\mathbf{S}_n$ sampled from $\mathbf{P}_n$, for any $\nu > 0$, assume $n$ is large enough such that $n^{-\nu} \leq \epsilon_b$, there exists a constant $C_\nu$ such that*

$$\|\mathbf{S}_n - \mathbf{P}_n\|_{2,2} \leq C_\nu \sqrt{n} \tag{56}$$

*with probability at least $1 - \epsilon_b$.*

**Lemma.** *(Convergence to Graphon)*

*Consider the sparse graphon $W_m$ generated from $W_{\mathbb{R}_+}$ that satisfies AS1, and the induced graphon $\overline{W}_{m,n}$ sampled from $W_m$. For any $\nu > 0$, assume $n$ is large enough such that $n^{-\nu} \leq \rho$, there exists a constant $C_\nu$ such that*

$$\Delta T_{W_n} \leq \frac{C_\nu}{\sqrt{n}} + \frac{2A_W t_m}{\sqrt{6n}\epsilon_u} \tag{57}$$

*with probability at least $(1 - \epsilon_u)(1 - \rho)$.*

*Proof.* Based on the triangle inequality, $\Delta T_{W_n}$ can be divided into two parts:

$$\Delta T_{W_n} \leq \left\| W - \overline{P}_n \right\|_{2,2} + \left\| \overline{P}_n - \overline{W}_n \right\|_{2,2}. \tag{58}$$

For $\left\| W - \overline{P}_n \right\|_{2,2}$, using Lemma 1 in Ji et al. (2024) without stretching, we have

$$\left\| W - \overline{P}_n \right\|_{2,2} \leq \left\| W - \overline{P}_n \right\|_2. \tag{59}$$

Similar to the proof of Lemma E.1.2, we first divide $\|\overline{P}_n - W\|_2$ into $n^2$ parts, then employ the $t_m A_W$-Lipschitz property, it becomes

$$\begin{aligned}
\left\| \overline{P}_n - W \right\|_2^2 &= \sum_{i=1}^n \sum_{j=1}^n \int\!\!\int_{\mathbb{I}_i \times \mathbb{I}_j} (W(u_i, u_j) - W(u, v))^2 \, du \, dv \\
&\leq A_W^2 t_m^2 \sum_{i=1}^n \sum_{j=1}^n \int\!\!\int_{\mathbb{I}_i \times \mathbb{I}_j} (|u_i - u| + |u_j - v|)^2 \, du \, dv \\
&\leq A_W^2 t_m^2 \sum_{i=1}^n \sum_{j=1}^n \int\!\!\int_{\mathbb{I}_i \times \mathbb{I}_j} 2(u_i - u)^2 + 2(u_j - v)^2 \, du \, dv \\
&= 4 A_W^2 t_m^2 \sum_{i=1}^n \int_{\mathbb{I}_i} (u_i - u)^2 \, du.
\end{aligned} \tag{60}$$

Using the same conclusion as in the proof of Lemma E.1.2, we have $\mathbb{E}\left[\left\| \overline{P}_n - W \right\|_{2,2}\right] \leq \frac{2A_W t_m}{\sqrt{6n}}$. Then with probability at least $(1 - \epsilon_u)$, it holds

$$\left\| \overline{P}_n - W \right\|_{2,2} \leq \frac{2A_W t_m}{\sqrt{6n}\epsilon_u}. \tag{61}$$

For $\left\| \overline{P}_n - \overline{W}_n \right\|_{2,2}$, combining Lemma 5 and Lemma 6, then for any $\nu > 0$, assume $n$ is large enough such that $n^{-\nu} \leq \rho$, there exists a constant $C_\nu$ such that

$$\left\| \overline{P}_n - \overline{W}_n \right\|_{2,2} \leq \frac{C_\nu}{\sqrt{n}}, \tag{62}$$

with probability at least $1 - \epsilon_b$. □

### E.2 PROOF OF LEMMA 3: STRETCHED FORM

#### E.2.1 BOUND OF STRETCHING COEFFICIENTS

**Lemma 7.** *(Convergence of Stretching Coefficients)*

*For the $L^1$-norm of graphon $W_m$ and the induced graphon $\overline{W}_{m,n}$, it holds*

$$\left| \|W_m\|_1 - \|\overline{W}_{m,n}\|_1 \right| \leq \Delta T_{W_n}. \tag{63}$$

*Proof.*

$$
\begin{aligned}
\left| |\overline{W}_n|_1 - |W|_1 \right| &= \left| \int\int_{[0,1]^2} \overline{W}_n \, dudv - \int\int_{[0,1]^2} W \, dudv \right| \\
&= \left| \int_{[0,1]} \int_{[0,1]} \left( \overline{W}_n - W \right) \mathbf{1}(u) du \, \mathbf{1}(v) dv \right| \\
&\leq \left\| \int_{[0,1]} \left( \overline{W}_n - W \right) \mathbf{1}(u) du \right\|_2 \left\| \mathbf{1}(v) \right\|_2 \\
&\leq \left\| \overline{W}_n - W \right\|_{2,2} \left\| \mathbf{1}(u) \right\|_2 \\
&= \left\| \overline{W}_n - W \right\|_{2,2} .
\end{aligned}
\tag{64}
$$

The first inequality is derived from the Cauchy-Schwarz inequality, taking $\int_{[0,1]} \left( \overline{W}_n - W \right) \mathbf{1}(u) du$ as a function of $v$, and the second inequality is based on the definition of the operator norm. $\qquad\square$

That is $\left| s^2 - s_n^2 \right| \leq \Delta T_{W_n}$ for stretching coefficients. When $n$ is sufficiently such that $\Delta T_{W_n} \leq s^2/2$, we have

$$
\frac{1}{s_n^2} \leq \frac{2}{s^2} .
\tag{65}
$$

### E.2.2 Stretching Effects on Graphon and Signal

**Lemma 8.** *Let* $(\mathcal{W}^{s_\forall}, X^{s_\forall})$ *be the stretched graphon and graphon signal of value-unbounded classical graphon* $\mathcal{W}$ *and classical graphon signal* $X$, *then it holds*

$$
\begin{aligned}
\|X^{s_\forall}\|_2 &= \frac{1}{\sqrt{s_\forall}} \|X\|_2 , \\
\|\mathcal{W}^{s_\forall}\|_{2,2} &= \frac{1}{s_\forall} \|\mathcal{W}\|_{2,2}
\end{aligned}
\tag{66}
$$

*with non-specific stretching coefficient* $s_\forall \in (0,1]$.

*Proof.* For stretched graphon signal $X^{s_\forall}$ with non-specific coefficient, consider its $L^2$-norm

$$
\begin{aligned}
\|X^{s_\forall}\|_2^2 &= \int_{[0,+\infty)} \left( X^{s_\forall}(u) \right)^2 du = \int_{[0,1/s_\forall]} \left( X(s_\forall u) \right)^2 du \\
&= \frac{1}{s_\forall} \int_{[0,1]} \left( X(u) \right)^2 du = \frac{1}{s_\forall} \|X\|_2^2 .
\end{aligned}
\tag{67}
$$

For stretched value-unbounded graphon $\mathcal{W}^{s_\forall}$, since it has all-zero regions where $u > 1/s_\forall$ or $v > 1/s_\forall$, for any $X_{\mathbb{R}_+} \in L^2(\mathbb{R}_+)$, we can find a corresponding stretched signal satisfying $X^{s_\forall}(u) = X(u)$ on $u \leq 1/s_\forall$ such that

$$
\left\| T_{\mathcal{W}^{s_\forall}} X_{\mathbb{R}_+} \right\|_2^2 = \left\| T_{\mathcal{W}^{s_\forall}} X^{s_\forall} \right\|_2^2 , \quad \left\| X_{\mathbb{R}_+} \right\|_2^2 \geq \left\| X^{s_\forall} \right\|_2^2 .
\tag{68}
$$

Therefore, the operator norm can be transformed to

$$
\begin{aligned}
\|\mathcal{W}^{s_\forall}\|_{2,2}^2 &= \sup_{\forall X_{\mathbb{R}_+} \in L^2(\mathbb{R}_+)} \frac{\left\| T_{\mathcal{W}^{s_\forall}} X_{\mathbb{R}_+} \right\|_2^2}{\left\| X_{\mathbb{R}_+} \right\|_2^2} = \sup_{\forall X^{s_\forall} \in L^2(\mathbb{R}_+)} \frac{\left\| T_{\mathcal{W}^{s_\forall}} X^{s_\forall} \right\|_2^2}{\left\| X^{s_\forall} \right\|_2^2} \\
&= \sup_{\forall X \in L^2([0,1])} \frac{\left\| T_{\mathcal{W}^{s_\forall}} X^{s_\forall} \right\|_2^2}{\left\| X^{s_\forall} \right\|_2^2} .
\end{aligned}
\tag{69}
$$

Although $X^{s_\forall}$ is constructed from $X \in L^2([0,1])$, the function class $X^{s_\forall}$ lies in $L^2([0,+\infty))$ can still represent all square-integrable functions supported on the interval $[0, 1/s_\forall]$.

For the numerator, we consider the following:

$$\|T_{\mathcal{W}^{s_\forall}} X^{s_\forall}\|_2^2 = \int_{[0,1/s_\forall]} \left( \int_{[0,1/s_\forall]} \mathcal{W}^{s_\forall}(u,v) X^{s_\forall}(u) du \right)^2 dv$$

$$= \left( \frac{1}{s_\forall} \right)^3 \int_{[0,1]} \left( \int_{[0,1]} \mathcal{W}(u,v) X(u) du \right)^2 dv \qquad (70)$$

$$= \left( \frac{1}{s_\forall} \right)^3 \|T_{\mathcal{W}} X\|_2^2.$$

For the denominator, we have $\|X^{s_\forall}\|_2^2 = \frac{1}{s_\forall} \|X\|_2^2$. Substituting the above results into Eq.( 69), we obtain

$$\|\mathcal{W}^{s_\forall}\|_{2,2}^2 = \frac{1}{s_\forall^2} \sup_{\forall X \in L^2([0,1])} \frac{\|T_{\mathcal{W}} X\|_2^2}{\|X\|_2^2} = \frac{1}{s_\forall^2} \|\mathcal{W}\|_{2,2}^2. \qquad (71)$$

$$\square$$

**Lemma 9.** *Consider different stretching coefficients $s_0, s_1 \in (0,1]$, applied to the same classical graphon $W$ with signal $X$, respectively. Assume $\left| s_1^2 - s_0^2 \right| \le s_0^2/2$ ($s_0, s_1$ corresponding to $s_m, s_{m,n}$), then it holds*

$$\|X^{s_0} - X^{s_1}\|_2 = C_{X,1} \left| s_0^2 - s_1^2 \right| + C_{X,2} \left| s_0^2 - s_1^2 \right|^{1/2},$$
$$\|W^{s_0} - W^{s_1}\|_{2,2} = C_{W,1} \left| s_0^2 - s_1^2 \right| + C_{W,2} \left| s_0^2 - s_1^2 \right|^{1/2}, \qquad (72)$$

*where $C_{X,1}, C_{X,2}$ and $C_{W,1}, C_{W,2}$ are constants about graphon signal $X$ and graphon $W$, described in the following proof.*

*Proof.* Apply different non-specific stretching coefficients $s_0, s_1 \in (0,1]$ to the same graphon signal $X_m$, which satisfies $A_X t_m$-Lipschitz. Using Lemma 8, the difference between $X^{s_0}$ and $X^{s_1}$ becomes

$$\|X^{s_0} - X^{s_1}\|_2 = \frac{1}{\sqrt{s_0}} \left\| X - X^{s'} \right\|_2, \qquad (73)$$

where $X^{s'}$ is stretched by the coefficient $s' = s_1/s_0$, that is $X^{s'}(u) = X(s'u)$ for $u \in [0, 1/s']$.

To compare the difference between $X \in L^2([0,1]^2)$ and $X^{s'}$ supported on $[0, 1/s']$, we need to discuss the effective integral area case-by-case:

$$\begin{cases} \textbf{Case 1:} & 1/s' \le 1, \\ \textbf{Case 2:} & 1/s' \ge 1. \end{cases} \qquad (74)$$

**Case 1:** When $1/s' \le 1$, $\left\| X - X^{s'} \right\|_2$ expands to

$$\left\| X - X^{s'} \right\|_2^2 = \underbrace{\int_{[0,\frac{1}{s'}]} \left( X(u) - X^{s'}(u) \right)^2 du}_{\textbf{(i)}} + \underbrace{\int_{[\frac{1}{s'},1]} X^2(u) du}_{\textbf{(ii)}}. \qquad (75)$$

The first part **(i)** represents the signal value difference, while the second part **(ii)** captures the area difference.

For the first part **(i)**, using the Lipschitz-property, we have

$$\textbf{(i)} = \int_{[0,\frac{1}{s'}]} \left( X(u) - X(s'u) \right)^2 du$$

$$\le \int_{[0,\frac{1}{s'}]} A_X^2 t_m^2 \left( u - s'u \right)^2 du = \frac{A_X^2 t_m^2 (1-s')^2}{3s'^3}. \qquad (76)$$

For the second part **(ii)**, the values of $X^2$ in $[\frac{1}{s'}, 1]$ is bounded by $\max(X^2)$. Therefore, we can directly calculate the area

$$\textbf{(ii)} = \int_{[\frac{1}{s'}, 1]} X^2(u) du \leq \max(X^2)\left(1 - \frac{1}{s'}\right). \tag{77}$$

Combining the results of **(i)** and **(ii)**, we have

$$\begin{aligned}\left\|X - X^{\mathfrak{s}'}\right\|_2^2 &\leq \frac{A_X^2 t_m^2 (1-s')^2}{3s'^3} + \max(X^2)\left(1 - \frac{1}{s'}\right) \\ &\leq \frac{A_X^2 t_m^2 (1-s')^2}{3} + \max(X^2)\left|1 - \frac{1}{s'}\right|,\end{aligned} \tag{78}$$

where the second inequality is derived by $1/s' \leq 1$.

**Case 2:** When $1/s' \geq 1$, $\left\|X - X^{\mathfrak{s}'}\right\|_2$ expands to

$$\left\|X - X^{\mathfrak{s}'}\right\|_2^2 = \underbrace{\int_{[0,1]} \left(X(u) - X^{\mathfrak{s}'}(u)\right)^2 du}_{\textbf{(i)}} + \underbrace{\int_{[1, \frac{1}{s'}]} X^{\mathfrak{s}'^2}(u) du}_{\textbf{(ii)}}. \tag{79}$$

Following a similar process to the above, we have

$$\begin{aligned}\textbf{(i)} &\leq \int_{[0,1]} A_X^2 t_m^2 (u - s'u)^2 \, du = \frac{A_X^2 t_m^2 (1-s')^2}{3}, \\ \textbf{(ii)} &\leq \max(X^2)\left(\frac{1}{s'} - 1\right) = \max(X^2)\left|\frac{1}{s'} - 1\right|.\end{aligned} \tag{80}$$

Combining the above two inequalities, we obtain

$$\left\|X - X^{\mathfrak{s}'}\right\|_2^2 \leq \frac{A_X^2 t_m^2 (1-s')^2}{3} + \max(X^2)\left|\frac{1}{s'} - 1\right|, \tag{81}$$

The results of **case 1** and **case 2** share the same form, then we prove the final conclusion

$$\begin{aligned}\left\|X^{\mathfrak{s}_0} - X^{\mathfrak{s}_1}\right\|_2 &= \frac{1}{\sqrt{s_0}} \left\|X - X^{\mathfrak{s}'}\right\|_2 \\ &\leq \frac{1}{\sqrt{s_0}} \left(\frac{A_X t_m}{\sqrt{3}} \left|1 - \frac{s_1}{s_0}\right| + \max(|X|)\left|1 - \frac{s_0}{s_1}\right|^{1/2}\right) \\ &= \frac{A_X t_m}{\sqrt{3} s_0^{3/2}(s_0 + s_1)} \left|s_0^2 - s_1^2\right| + \frac{\max(|X|)}{\sqrt{s_0 s_1 (s_0 + s_1)}} \left|s_0^2 - s_1^2\right|^{1/2}.\end{aligned} \tag{82}$$

When $\left|s_1^2 - s_0^2\right| \leq s_0^2/2$ for stretching coefficients, we have $\frac{1}{s_1^2} \leq \frac{2}{s_0^2}$. The above inequality becomes

$$\begin{aligned}\left\|X^{\mathfrak{s}_0} - X^{\mathfrak{s}_1}\right\|_2 &\leq \frac{A_X t_m}{\sqrt{3} s_0^{3/2}(s_0 + s_1)} \left|s_0^2 - s_1^2\right| + \frac{\max(|X|)}{\sqrt{s_0 s_1 (s_0 + s_1)}} \left|s_0^2 - s_1^2\right|^{1/2} \\ &\leq C_{X,1}\left|s_0^2 - s_1^2\right| + C_{X,2}\left|s_0^2 - s_1^2\right|^{1/2},\end{aligned} \tag{83}$$

where $C_{X,1} = \frac{A_X t_m}{\sqrt{3} s_0^{5/2}(1+1/\sqrt{2})}, C_{X,2} = \frac{\max(|X|)}{\sqrt{s_0^{5/2}/\sqrt{2}(1+1/\sqrt{2})}}$ are constans about $X$ and the stretching coefficient $s_0$.

Apply the two different non-specific stretching coefficients $s_0, s_1 \in (0,1]$ to the same graphon $W$, which satisfies $A_W t_m$-Lipschitz, then consider the difference $\|W^{\mathfrak{s}_0} - W^{\mathfrak{s}_1}\|_{2,2}$, by Lemma 8 and Lemma 1 in Ji et al. (2024), it becomes

$$\|W^{\mathfrak{s}_0} - W^{\mathfrak{s}_1}\|_{2,2} \leq \frac{1}{s_0} \left\|W - W^{\mathfrak{s}'}\right\|_2, \tag{84}$$

where $W^{\mathsf{s}'}$ is stretched based on the coefficient $s' = s_1/s_0$, that is $W^{\mathsf{s}'}(u,v) = W(s'u, s'v)$ for $(u,v) \in [0, 1/s']^2$. Similarly, we analyze the difference under two distinct cases: $1/s' \leq 1$ or $1/s' \geq 1$.

**Case 1:** When $1/s' \leq 1$, $\left\| W - W^{\mathsf{s}'} \right\|_2$ expands to

$$\left\| W - W^{\mathsf{s}'} \right\|_2^2 = \underbrace{\int\int_{[0, \frac{1}{s'}]^2} \left( W(u,v) - W^{\mathsf{s}'}(u,v) \right)^2 dudv)}_{\textbf{(i)}}$$
$$+ \underbrace{\int\int_{[0,1]^2 - [0, \frac{1}{s'}]^2} W^2(u,v) dudv}_{\textbf{(ii)}} . \tag{85}$$

The value of $W$ in **(ii)** is bounded by 1. Following the same process used previously, we obtain

$$\textbf{(i)} \leq A_W^2 t_m^2 (1-s')^2 \int\int_{[0, \frac{1}{s'}]^2} (u+v)^2 \, dudv = \frac{7 A_W^2 t_m^2 (1-s')^2}{6 s'^4} .$$

$$\textbf{(ii)} \leq \int\int_{[0,1]^2 - [0, \frac{1}{s'}]^2} 1 dudv = 1 - \frac{1}{s'^2} . \tag{86}$$

Combining the above, we have the following:

$$\textbf{(i)} + \textbf{(ii)} \leq \frac{7 A_W^2 t_m^2 (1-s')^2}{6 s'^4} + \left( 1 - \frac{1}{s'^2} \right)$$
$$\leq \frac{7 A_W^2 t_m^2 (1-s')^2}{6} + \left| 1 - \frac{1}{s'^2} \right| , \tag{87}$$

where the second inequality is derived by $1/s' \leq 1$.

**Case 2:** When $1/s' \geq 1$, $\left\| W - W^{\mathsf{s}'} \right\|_2$ expands to

$$\left\| W - W^{\mathsf{s}'} \right\|_2^2 = \underbrace{\int\int_{[0,1]^2} \left( W(u,v) - W^{\mathsf{s}'}(u,v) \right)^2 dudv}_{\textbf{(i)}}$$
$$+ \underbrace{\int\int_{[0, \frac{1}{s'}]^2 - [0,1]^2} W^{\mathsf{s}'^2}(u,v) dudv}_{\textbf{(ii)}} . \tag{88}$$

Similarly, the value of $W^{\mathsf{s}'}$ is bounded by $A_W(1-s')$. Then use the Lipschitz property to calculate **(i)**, and use the value bound to calculate **(ii)**:

$$\textbf{(i)} \leq A_W^2 t_m^2 (1-s')^2 \int\int_{[0,1]^2} (u+v)^2 \, dudv = \frac{7 A_W^2 t_m^2 (1-s')^2}{6} .$$

$$\textbf{(ii)} \leq \int\int_{[0, \frac{1}{s'}]^2 - [0,1]^2} 1 dudv = \frac{1}{s'^2} - 1 . \tag{89}$$

Combining the above, we have the following.

$$\textbf{(i)} + \textbf{(ii)} \leq \frac{7 A_W^2 t_m^2 (1-s')^2}{6} + \left| \frac{1}{s'^2} - 1 \right| . \tag{90}$$

Since results of **case 1** and **case 2** share the same form, we obtain

$$\left\| W^{\mathsf{s}_0} - W^{\mathsf{s}_1} \right\|_{2,2} \leq \frac{1}{s_0} \left\| W - W^{\mathsf{s}'} \right\|_2$$
$$\leq \frac{1}{s_0} \left( \sqrt{\frac{7 A_W^2 t_m^2}{6}} |1 - s'| + \left| 1 - \frac{1}{s'^2} \right|^{1/2} \right) \tag{91}$$
$$\leq \frac{\sqrt{7} A_W t_m}{\sqrt{6} s_0^2 (s_0 + s_1)} |s_0^2 - s_1^2| + \frac{1}{s_0 \sqrt{s_1(s_0 + s_1)}} |s_0^2 - s_1^2|^{1/2} .$$

When $\left|s_1^2 - s_0^2\right| \le s_0^2/2$ for stretching coefficients, we have $\frac{1}{s_1^2} \le \frac{2}{s_0^2}$. The above inequality becomes

$$\left\|W^{\mathfrak{s}_0} - W^{\mathfrak{s}_1}\right\|_{2,2} \le C_{W,1}\left|s_0^2 - s_1^2\right| + C_{W,2}\left|s_0^2 - s_1^2\right|^{1/2}, \tag{92}$$

where $C_{W,1} = \frac{\sqrt{7}A_W t_m}{\sqrt{6}s_0^3(1+1/\sqrt{2})}, C_{W,2} = \frac{1}{s_0^2\sqrt{(1+1/\sqrt{2})/\sqrt{2}}}$ are constants about $W$ and the stretching coefficient $s_0$. $\qquad\square$

### E.2.3 STRETCHED GRAPH SEQUENCE CONVERGENCE (PROOF OF LEMMA 3)

For better distinction, we use $s$ and $s_n$ to denote different stretching coefficients of $W$ and $\overline{W}_n$. The differences of stretched graphons and stretched signals are derived by

$$\left\|W^{\mathfrak{s}} - \overline{W}_n^{\mathfrak{s}_n}\right\|_{2,2} \le \underbrace{\left\|W^{\mathfrak{s}} - W^{\mathfrak{s}_n}\right\|_{2,2}}_{\textbf{(a)}} + \underbrace{\left\|W^{\mathfrak{s}_n} - \overline{W}_n^{\mathfrak{s}_n}\right\|_{2,2}}_{\textbf{(b)}},$$

$$\left\|X^{\mathfrak{s}} - \overline{X}_n^{\mathfrak{s}}\right\|_2 \le \underbrace{\left\|X^{\mathfrak{s}} - X^{\mathfrak{s}_n}\right\|_2}_{\textbf{(a)}} + \underbrace{\left\|X^{\mathfrak{s}_n} - \overline{X}_n^{\mathfrak{s}_n}\right\|_2}_{\textbf{(b)}}, \tag{93}$$

where $W^{\mathfrak{s}_n}$ and $X^{\mathfrak{s}_n}$ are generalized stretched forms of $W$ and $X$ with coefficient $s_n$. Assume $n$ is sufficiently large such that $\left|s^2 - s_n^2\right| \le s^2/2$ in Lemma 7, then applying the above Lemma 8 and Lemma 9, we obtain

$$\left\|W^{\mathfrak{s}} - \overline{W}_n^{\mathfrak{s}_n}\right\|_{2,2} \le \underbrace{C_{W,1}\left|s_0^2 - s_1^2\right| + C_{W,2}\left|s_0^2 - s_1^2\right|^{1/2}}_{\textbf{(a)}} + \underbrace{\frac{1}{s_n}\Delta T_{W_n}}_{\textbf{(b)}}$$

$$\le C_{W,1}\Delta T_{W_n} + C_{W,2}\Delta T_{W_n}^{1/2} + \frac{\sqrt{2}}{s}\Delta T_{W_n},$$

$$\left\|X^{\mathfrak{s}} - \overline{X}_n^{\mathfrak{s}}\right\|_2 \le \underbrace{C_{X,1}\left|s_0^2 - s_1^2\right| + C_{X,2}\left|s_0^2 - s_1^2\right|^{1/2}}_{\textbf{(a)}} + \underbrace{\frac{1}{\sqrt{s_n}}\Delta X_n}_{\textbf{(b)}} \tag{94}$$

$$\le C_{X,1}\Delta T_{W_n} + C_{X,2}\Delta T_{W_n}^{1/2} + \frac{2^{\frac{1}{4}}}{\sqrt{s}}\Delta X_n.$$

For simplicity, we leverage $C_{\mathfrak{s}}$ to represent the larger value of $\frac{\sqrt{2}}{s}$ and $\frac{2^{\frac{1}{4}}}{\sqrt{s}}$, that is

$$\left\|W^{\mathfrak{s}} - \overline{W}_n^{\mathfrak{s}_n}\right\|_{2,2} \le C_{\mathfrak{s}}\Delta T_{W_n} + C_{W,\mathfrak{s}}\Delta T_{W_n}^{1/2},$$

$$\left\|X^{\mathfrak{s}} - \overline{X}_n^{\mathfrak{s}}\right\|_2 \le C_{\mathfrak{s}}\Delta X_n + C_{X,\mathfrak{s}}\Delta T_{W_n}^{1/2}, \tag{95}$$

where $C_{\mathfrak{s}} := \max\{\frac{\sqrt{2}}{s}, \frac{2^{\frac{1}{4}}}{\sqrt{s}}\}$ is constant about $W$, $C_{W,\mathfrak{s}} = C_{W,1}\Delta T_{W_n}^{1/2} + C_{W,2}$ and $C_{X,\mathfrak{s}} = C_{X,1}\Delta T_{W_n}^{1/2} + C_{X,2}$ converge to constants $C_{W,2}$ and $C_{X,2}$ as $n$ increases.

## F PROOF OF LEMMA 1 AND THEOREM 1 2: FILTER AND SWNN

Since we have already established the convergence of sparse graph models with respect to both the stretched graphon and the stretched signal, it follows that when the difference between filters (or SWNN outputs) is bounded by the differences of the underlying graphons and signals, the convergence of the filters and the SWNN naturally holds as well.

### F.1 GRAPH CONVOLUTIONAL NETWORKS

Through the neighborhood structure defined by the graph topology, graph convolution allows each node to aggregate the features of its neighbors. As a signal processing framework on graphs, a graph

convolutional operator (Segarra et al., 2017; Du et al., 2018; Gama et al., 2019) of $R$ order is defined as

$$h(\mathbf{S}_n)\boldsymbol{x}_n = \sum_{r=0}^{R} h_r \mathbf{S}_n^r \boldsymbol{x}_n, \tag{96}$$

where coefficients $h_r$ are weights of the $r$-th order convolution, aggregating features of each node's $r$-hop neighborhood.

Let $\Phi(\mathbf{S}_n, \boldsymbol{x}_n, \mathcal{H})$ denote GCN with learnable parameters $\mathcal{H}$, and 1-dimension input features $\boldsymbol{x}_n$ for simplicity. For an $L$-layers GCN, we denote the input/output feature dimension of each layer $l$ ($1 \leq l \leq L$) by $F^{l-1}/F^l$, with corresponding parameters $\mathcal{H}^l \in \mathbb{R}^{F^{l-1} \times F^l}$, which implies the relationship between input features and output features. For each layer $l$, the aggregation and propagation process can be expressed in channel-wise form (Ruiz et al., 2020; 2021b; 2023):

$$\boldsymbol{x}_{f_l} = \sigma \left( \sum_{f_{l-1}=1}^{F_{l-1}} h_{f_{l-1}, f_l}(\mathbf{S}_n) \boldsymbol{x}_{f_{l-1}} \right). \tag{97}$$

The activation function is denoted as $\sigma(\cdot)$. The learnable parameters $\mathcal{H}^l(f_{l-1}, f_l)$ are maintained in the weights of convolutional operators $h_{f_{l-1}, f_l}(\cdot) = \mathcal{H}^l(f_{l-1}, f_l)h(\cdot)$ in Eq.( 96), which is fixed during convergence and transferability.

## F.2 DIFFERENCE BOUND OF FILTER OUTPUTS

**Lemma 10.** *(Difference Bound of Filter Outputs)*

*Let $h(\cdot)$ denote the convolutional filter, applied to stretched graphons with signals $(W_1^{\mathfrak{s}}, X_1^{\mathfrak{s}})$ and $(W_1^{\mathfrak{s}}, X_1^{\mathfrak{s}})$, it holds*

$$\|h(W_1^{\mathfrak{s}})X_1^{\mathfrak{s}} - h(W_2^{\mathfrak{s}})X_2^{\mathfrak{s}}\|_2 \leq C_h \left( \Delta T_{W^{\mathfrak{s}}} \|X_1^{\mathfrak{s}}\|_2 + \Delta X^{\mathfrak{s}} \right), \tag{98}$$

*where $C_h$ is a constant related to the filter $h(\cdot)$ described in F.2.*

*Proof.* (**Proof of Lemma 10**.)

Using the triangle inequality, the filter convergence error is divided into two parts:

$$\|h(W_1^{\mathfrak{s}})X_1^{\mathfrak{s}} - h(W_2^{\mathfrak{s}})X_2^{\mathfrak{s}}\|_2$$
$$\leq \underbrace{\|h(W_1^{\mathfrak{s}}) - h(W_2^{\mathfrak{s}})\|_{2,2} \|X_1^{\mathfrak{s}}\|_2}_{\textbf{(a)}} + \underbrace{\|h(W_2^{\mathfrak{s}})\|_{2,2} \Delta X^{\mathfrak{s}}}_{\textbf{(b)}}. \tag{99}$$

Parts **(a)** and **(b)** represent the operator difference and the signal difference, respectively.

Before proceeding to further analysis, we first consider the simplest case, namely the 1-order shift operator $T_{W^{\mathfrak{s}}}$. For any stretched graphon $W^{\mathfrak{s}}$, its $L^1$-norm is equal to 1, i.e. $\|W^{\mathfrak{s}}\|_1 = 1$, then we have

$$\|W^{\mathfrak{s}}\|_{2,2}^2 \leq \int \int_{[0,+\infty)^2} (W^{\mathfrak{s}})^2 du dv$$
$$\leq \max(W^{\mathfrak{s}}) \int \int_{[0,+\infty)^2} W^{\mathfrak{s}} du dv \tag{100}$$
$$= \|W^{\mathfrak{s}}\|_1 = 1,$$

the first inequality is derived by Lemma 1 in Ji et al. (2024). Recursively, we can derive the bound for the $r$-order operator, which is also valid for $W_2^{\mathfrak{s}}$, that is

$$\left\| T_{W^{\mathfrak{s}}}^{(r)} \right\|_{2,2} \leq 1. \tag{101}$$

For part **(a)**, we expand of the difference operator norm:

$$\|h(W_1^{\mathfrak{s}}) - h(W_2^{\mathfrak{s}})\|_{2,2} = \left\| \sum_{r=0}^{R} h_r \left( T_{W_1^{\mathfrak{s}}}^{(r)} - T_{W_2^{\mathfrak{s}}}^{(r)} \right) \right\|_{2,2}$$
$$\leq \sum_{r=0}^{R} h_r \left\| T_{W_1^{\mathfrak{s}}}^{(r)} - T_{W_2^{\mathfrak{s}}}^{(r)} \right\|_{2,2}, \tag{102}$$

the Cauchy-Schwarz inequality is used to derive the inequality. For the $r$-order shift operator, we have

$$
\begin{aligned}
& \left\| T_{W_1^{\mathfrak{s}}}^{(r)} - T_{W_2^{\mathfrak{s}}}^{(r)} \right\|_{2,2} \\
&= \left\| T_{W_1^{\mathfrak{s}}} \left( T_{W_1^{\mathfrak{s}}}^{(r-1)} - T_{W_2^{\mathfrak{s}}}^{(r-1)} \right) - \left( T_{W_1^{\mathfrak{s}}} - T_{W_2^{\mathfrak{s}}} \right) T_{W_2^{\mathfrak{s}}}^{(r-1)} \right\|_{2,2} \\
&\leq \left\| T_{W_1^{\mathfrak{s}}}^{(r-1)} - T_{W_2^{\mathfrak{s}}}^{(r-1)} \right\|_{2,2} + \left\| T_{W_1^{\mathfrak{s}}} - T_{W_2^{\mathfrak{s}}} \right\|_{2,2} \\
&\leq r \left\| T_{W_1^{\mathfrak{s}}} - T_{W_2^{\mathfrak{s}}} \right\|_{2,2}.
\end{aligned}
\tag{103}
$$

Substituting the above into the expression( 102), we obtain

$$
\| h(W_1^{\mathfrak{s}}) - h(W_2^{\mathfrak{s}}) \|_{2,2} \leq \left( \sum_{r=0}^{R} h_r r \right) \Delta T_{W^{\mathfrak{s}}}.
\tag{104}
$$

For part **(b)**, based on the properties of inequality( 101), we can derive:

$$
\| h(W_2^{\mathfrak{s}}) \|_{2,2} \leq \sum_{r=0}^{R} h_r \left\| T_{W_2^{\mathfrak{s}}}^{(r)} \right\|_{2,2} \leq \sum_{r=0}^{R} h_r.
\tag{105}
$$

By setting the constant $C_h := \max(\sum_{r=0}^{R} h_r r, \sum_{r=0}^{R} h_r)$, we complete the proof.

$\square$

### F.3 Difference Bound of SWNN Outputs

*Proof.* To distinguish between signals from different layers, we denote the output features of the $l$-th layer by $X_{f_L,1}^{\mathfrak{s}}$ and $X_{f_L,2}^{\mathfrak{s}}$, belonging to $\Phi(W_1^{\mathfrak{s}}, X_1^{\mathfrak{s}}, \mathcal{H})$ and $\Phi(W_2^{\mathfrak{s}}, X_2^{\mathfrak{s}}, \mathcal{H})$ respectively.

We analyze the SWNNs difference $\Delta \Phi^{\mathfrak{s}}$ from the last layer,

$$
(\Delta \Phi^{\mathfrak{s}})^2 = \sum_{f_L=1}^{F_L} \left\| X_{f_L,1}^{\mathfrak{s}} - X_{f_L,2}^{\mathfrak{s}} \right\|_2^2.
\tag{106}
$$

According to the aggregation and propagation process of SWNNs, the output features can be expanded as

$$
\begin{aligned}
X_{f_L,1}^{\mathfrak{s}} &= \sigma \left( \sum_{f_{L-1}}^{F_{L-1}} h_{f_{L-1},f_L}(W_1^{\mathfrak{s}}) X_{f_{L-1},1}^{\mathfrak{s}} \right), \\
X_{f_L,2}^{\mathfrak{s}} &= \sigma \left( \sum_{f_{L-1}}^{F_{L-1}} h_{f_{L-1},f_L}(W_2^{\mathfrak{s}}) X_{f_{L-1},2}^{\mathfrak{s}} \right).
\end{aligned}
\tag{107}
$$

Because the activation functions are normalized Lipschitz, that is $|\sigma(a) - \sigma(b)| \leq |a - b|$, we derive the above

$$
\begin{aligned}
\left\| X_{f_L,1}^{\mathfrak{s}} X_{f_L,2}^{\mathfrak{s}} \right\|_2 &\leq \left\| \sum_{f_{L-1}}^{F_{L-1}} h_{f_{L-1},f_L}(W_1^{\mathfrak{s}}) X_{f_{L-1},1}^{\mathfrak{s}} - \sum_{f_{L-1}}^{F_{L-1}} h_{f_{L-1},f_L}(W_2^{\mathfrak{s}}) X_{f_{L-1},2}^{\mathfrak{s}} \right\|_2 \\
&\leq \sum_{f_{L-1}}^{F_{L-1}} \left\| h_{f_{L-1},f_L}(W_1^{\mathfrak{s}}) X_{f_{L-1},1}^{\mathfrak{s}} - h_{f_{L-1},f_L}(W_2^{\mathfrak{s}}) X_{f_{L-1},2}^{\mathfrak{s}} \right\|_2 \\
&\leq \sum_{f_{L-1}}^{F_{L-1}} C_{\mathcal{H}} \left( \Delta T_{W^{\mathfrak{s}}} \left\| X_{f_{L-1},1}^{\mathfrak{s}} \right\|_2 + \left\| X_{f_{L-1},1}^{\mathfrak{s}} - X_{f_{L-1},2}^{\mathfrak{s}} \right\|_2 \right).
\end{aligned}
\tag{108}
$$

The second inequality is derived from the Cauchy-Schwarz inequality, and the third inequality is derived by Lemma 10. The constant $C_{\mathcal{H}}$ is the maximum of all channel-wise filters $C_{h_{f_{l-1},f_l}}$, which in turn depends on $\mathcal{H}$.

Using the assumption about the activation function $\sigma(\cdot)$, we get $|\sigma(x)| = |\sigma(x) - 0| = |\sigma(x) - \sigma(0)| \leq |x - 0| = |x|$, then $\|X^{\mathfrak{s}}_{f_{L-1},1}\|_2$ can be derived by

$$
\begin{aligned}
\left\| X^{\mathfrak{s}}_{f_{L-1},1} \right\|_2 &\leq \left\| \sum_{f_{L-2}}^{F_{L-2}} h_{f_{L-2},f_{L-1}}(W^{\mathfrak{s}}_1) X^{\mathfrak{s}}_{f_{L-2},1} \right\|_2 \\
&\leq \sum_{f_{L-2}}^{F_{L-2}} C_{\mathcal{H}} \left\| X^{\mathfrak{s}}_{f_{L-2},1} \right\|_2 \\
&\leq \prod_{l=1}^{L-2} C_{\mathcal{H}} F_l \sum_{f_0}^{F_0} \left\| X^{\mathfrak{s}}_{f_0,1} \right\|_2 \\
&\overset{F_0=1}{=} \prod_{l=0}^{L-2} C_{\mathcal{H}} F_l \left\| X^{\mathfrak{s}}_1 \right\|_2 .
\end{aligned}
\tag{109}
$$

The second inequality is derived by considering the operator norm of filters in Lemma 10, and the third inequality is obtained by calculating recursively. Substituting the inequality( 109) into the above result( 108), we have

$$
\left\| X^{\mathfrak{s}}_{f_L,1} - X^{\mathfrak{s}}_{f_L,2} \right\|_2 = \left( \prod_{l=0}^{L-1} C_{\mathcal{H}} F_l \right) \Delta T_{W^{\mathfrak{s}}} \left\| X^{\mathfrak{s}}_1 \right\|_2 + C_{\mathcal{H}} \sum_{f_{L-1}}^{F_{L-1}} \left\| X^{\mathfrak{s}}_{f_{L-1},1} - X^{\mathfrak{s}}_{f_{L-1},2} \right\|_2 , \tag{110}
$$

expanded recursively, we have

$$
\left\| X^{\mathfrak{s}}_{f_L,1} - X^{\mathfrak{s}}_{f_L,2} \right\|_2 \leq L \left( \prod_{l=0}^{L-1} C_{\mathcal{H}} F_l \right) \Delta T_{W^{\mathfrak{s}}_1} \left\| X^{\mathfrak{s}}_1 \right\|_2 + C_{\mathcal{H}}^L \sum_{f_0}^{F_0} \left\| X^{\mathfrak{s}}_{f_0,1} - X^{\mathfrak{s}}_{f_0,2} \right\|_2 . \tag{111}
$$

For simplicity, we have set $F_0 = 1$ and $F_L = 1$, then it holds

$$
\left\| \Phi(W^{\mathfrak{s}}_1, X^{\mathfrak{s}}, \mathcal{H}) - \Phi(W^{\mathfrak{s}}_2, X^{\mathfrak{s}}_2, \mathcal{H}) \right\|_2 \leq C(\mathcal{H}) \left( \Delta T_{W^{\mathfrak{s}}} \left\| X^{\mathfrak{s}}_1 \right\|_2 + \Delta X^{\mathfrak{s}} \right) , \tag{112}
$$

where $C(\mathcal{H}) := \max(C_1(\mathcal{H}), C_2(\mathcal{H}))$, $C_1(\mathcal{H}) = L \left( \prod_{l=0}^{L-1} C_{\mathcal{H}} F_l \right)$ and $C_2(\mathcal{H}) = C_{\mathcal{H}}^L$ are constants about the model parameters $\mathcal{H}$.

$\square$

### F.4 Convergence and Transferability of SWNNs

**Theorem 2** (Convergence of SWNNs). *Consider an L-layer SWNN with learned parameters $\mathcal{H}$, denoted by $\Phi(W^{\mathfrak{s}}, X^{\mathfrak{s}}, \mathcal{H})$, where $F_0 = F_1 = 1$ for simplicity. Under assumptions from AS1 to AS3, and conditions on truncation length $t_m$ and graph size $n$ in Lemma 2, Lemma 3, and Lemma 4, then for any $k' > 0$, it holds that*

$$
\Delta \Phi^{\mathfrak{s}}_{m,n} \leq C(\mathcal{H}) \left( \frac{C_{\mathbb{R}_+}}{t_m^{k'}} + \frac{\|X_{\mathbb{R}_+}\|_{L^2([t_m,+\infty))}}{\|W_{\mathbb{R}_+}\|_1^{1/4}} \right) + C(\mathcal{H}) \left( \frac{C_{m,1}}{n^{1/2}} + \frac{C_{m,2}}{n^{1/4}} \right) . \tag{113}
$$

*with probability at least $(1 - \epsilon_u)(1 - \epsilon_b)$. Here, $C(\mathcal{H})$ depends on $\mathcal{H}$, $C_{\mathbb{R}_+} = C_{k',W} \left\| X^{\mathfrak{s}}_{\mathbb{R}_+} \right\|_2 + C_{k',X}$, $C_{m,1} = C_{\mathfrak{s}} t_m (C_W \|X^{\mathfrak{s}}_m\|_2 + C_X)$, $C_{m,2} = (C_W t_m)^{1/2} (C_{W,\mathfrak{s}} \|X^{\mathfrak{s}}_m\|_2 + C_{X,\mathfrak{s}})$.*

Based on Theorem 2, we also prove Theorem 1 through a triangle inequality.

*Proof.* According to the sampling process of the sparse graph model, i.e. $W_{\mathbb{R}_+} \sim W_m \sim \overline{W}_{m,n}$, the same for $X_{\mathbb{R}_+} \sim X_m \sim \overline{X}_{m,n}$, the convergence error of graph sequences is divided into two parts:

**Overall Error:** $\Delta \Phi^{\mathfrak{s}}_{m,n} = \left\| \Phi(\overline{W}^{\mathfrak{s}}_{m,n}, \overline{X}^{\mathfrak{s}}_{m,n}, \mathcal{H}) - \Phi(W^{\mathfrak{s}}_{\mathbb{R}_+}, X^{\mathfrak{s}}_{\mathbb{R}_+}, \mathcal{H}) \right\|_2 ,$

$$
\begin{cases}
\Delta \Phi^{\mathfrak{s}}_m = \left\| \Phi(W^{\mathfrak{s}}_m, X^{\mathfrak{s}}_m, \mathcal{H}) - \Phi(W^{\mathfrak{s}}_{\mathbb{R}_+}, X^{\mathfrak{s}}_{\mathbb{R}_+}, \mathcal{H}) \right\|_2 , \\
\Delta \Phi^{\mathfrak{s}}_n = \left\| \Phi(\overline{W}^{\mathfrak{s}}_{m,n}, \overline{X}^{\mathfrak{s}}_{m,n}, \mathcal{H}) - \Phi(W^{\mathfrak{s}}_m, X^{\mathfrak{s}}_m, \mathcal{H}) \right\|_2 .
\end{cases}
\tag{114}
$$

These two parts are divided by the triangle inequality, that is $\Delta\Phi_{m,n}^{\mathfrak{s}} \leq \Delta\Phi_m^{\mathfrak{s}} + \Delta\Phi_n^{\mathfrak{s}}$. Then based on Lemma 1, we have

$$\Delta\Phi_{m,n}^{\mathfrak{s}} \leq C(\boldsymbol{\mathcal{H}})\left(\Delta T_{W_m^{\mathfrak{s}}}\left\|X_{\mathbb{R}_+}^{\mathfrak{s}}\right\|_2 + \Delta X_m^{\mathfrak{s}}\right) + C(\boldsymbol{\mathcal{H}})\left(\Delta T_{W_n^{\mathfrak{s}}}\|X_m^{\mathfrak{s}}\|_2 + \Delta X_n^{\mathfrak{s}}\right), \tag{115}$$

where $\Delta T_{W_m^{\mathfrak{s}}} = \left\|W_{\mathbb{R}_+}^{\mathfrak{s}} - W_m^{\mathfrak{s}}\right\|_{2,2}$ and $\Delta X_m^{\mathfrak{s}} = \left\|X_{\mathbb{R}_+}^{\mathfrak{s}} - X_m^{\mathfrak{s}}\right\|_2$. Let's review the above results of Lemma 2, Lemma 4 and Lemma 3:

$$\begin{cases} \Delta T_{W_m^{\mathfrak{s}}} \leq \dfrac{C_{k',W}}{t_m^{k'}}, \\ \Delta X_m^{\mathfrak{s}} \leq \dfrac{C_{k',X}}{t_m^{k'}} + \dfrac{\left\|X_{\mathbb{R}_+}\right\|_{L^2([t_m,+\infty))}}{\sqrt{s_R}}. \end{cases} \tag{116}$$

$$\begin{cases} \Delta T_{W_n^{\mathfrak{s}}} \leq C_{\mathfrak{s}}\Delta T_{W_n} + C_{W,\mathfrak{s}}\Delta T_{W_n}^{1/2}, \\ \Delta X_n^{\mathfrak{s}} \leq C_{\mathfrak{s}}\Delta X_n + C_{X,\mathfrak{s}}\Delta T_{W_n}^{1/2}. \end{cases} \qquad \begin{cases} \Delta T_{W_n} \leq \dfrac{C_W}{\sqrt{n}}t_m, \\ \Delta X_n \leq \dfrac{C_X}{\sqrt{n}}t_m. \end{cases}$$

Continuing the conditions and probabilities of these lemmas and substitute them into the above formula, we obtain

$$\Delta\Phi_{m,n}^{\mathfrak{s}} \leq C(\boldsymbol{\mathcal{H}})\left(\frac{C_{k',W}\left\|X_{\mathbb{R}_+}^{\mathfrak{s}}\right\|_2 + C_{k',X}}{t_m^{k'}} + \frac{\left\|X_{\mathbb{R}_+}\right\|_{L^2([t_m,+\infty))}}{\sqrt{s_R}}\right)$$

$$+ C(\boldsymbol{\mathcal{H}})\left(\frac{C_{\mathfrak{s}}t_m(C_W\|X_m^{\mathfrak{s}}\|_2 + C_X)}{n^{1/2}} + \frac{(C_W t_m)^{1/2}(C_{W,\mathfrak{s}}\|X_m^{\mathfrak{s}}\|_2 + C_{W,\mathfrak{s}})}{n^{1/4}}\right). \tag{117}$$

$\square$