# OpenReview forum: "Transfer Bound of Graph Convolutional Networks across Arbitrary Sparsity"
_ICLR.cc/2026/Conference — Submitted to ICLR 2026_

### Official Review · Reviewer_vGFE · 2025-10-28

**Soundness:** 2
**Presentation:** 3
**Contribution:** 2
**Rating:** 4
**Confidence:** 3

**Summary:**

This paper develops a unified theoretical framework for sparse Graph Convolutional Networks by introducing the "stretched graphon" model. Addressing the limitation that classical graphons vanish in sparse limits, this framework rescales sparse graphs to ensure convergence to a stable, non-zero representation. The key contribution is a Transferability Theorem proving that GCNs can transfer across arbitrary sparsity levels, extending prior work limited to fixed sparsity. The authors formally establish a double-convergence framework for asymptotic consistency and empirically validate their findings on the Cora dataset.

**Strengths:**

## Stretched Graphon Framework

The paper generalizes the classical dense graphon limit to a stretched graphon framework that remains non-degenerate for sparse graphs. This is a novel theoretical contribution that conceptually bridges two previously disconnected regimes, dense and sparse graphs, under a single mathematical limit. Although there are many mathematical definitions, the writing is dense but logically coherent, with intuitive illustrations.

## Formal Convergence Analysis

The paper provides formal convergence proofs for both sparse graphon sequences and random graph sequences, then integrates them to derive the transferability theorem. This framework can serve as a theoretical foundation for studying generalization, transfer, and stability of GNNs on sparse and heterogeneous networks.

**Weaknesses:**

## Experimental Scope Limitations

The experiments only use one dataset (Cora), which is a small citation network with 2,708 nodes and a fixed label set. Given the paper's theoretical ambition to establish transferability across arbitrary sparsity, validation on a single graph is insufficient. The authors should include larger and structurally diverse graphs such as PubMed, OGB datasets, or synthetic graphon-generated graphs to demonstrate that the observed convergence trends generalize beyond one topology. In particular, testing on graphs with heterogeneous degree distributions (power-law networks) or directed and weighted edges would better demonstrate the robustness of the stretching mechanism and support the universality claims.

## Limited Model Diversity and Ablation Studies

Currently, only standard GCNs with 2–3 layers and 32/64 hidden units are tested. Since SWNN is presented as a general limit model, it would be valuable to test whether GCN variants such as GraphSAGE and GAT also exhibit similar stretched transfer behavior. This would provide empirical confirmation of the framework's universality and show that the theoretical insights apply beyond the specific GCN architecture tested.

## Constructive Suggestions for Improvement

- Extend evaluation to multiple datasets and include synthetic sparse graphons for controlled experimentation
- Test robustness across different GNN architectures and normalization variants
- Include larger-scale datasets and graphs with diverse structural properties

## Overall Assessment

The paper's theoretical contribution is strong and original, but the experimental validation is narrow and only partially demonstrates the claimed universality. Expanding the empirical scope and quantitatively aligning results with theory would substantially strengthen the work.

**Questions:**

1. Could the same convergence and transferability arguments apply to transformer-style graph networks?
2. Could you elaborate more intuitively on why stretching by the L¹-norm specifically restores a non-zero limit for sparse graphons?

---

> ### Author Response · Authors · 2025-11-20
> **Response to Reviewer vGFE**
>
> We appreciate your suggestions on the experiments and your questions, which help us clarify the physical intuition of our framework.
>
> **>>>W1. Experimental Scope Limitations**
>
> **>>>R1:** We thank you for this valuable suggestion. Adopting the constructive suggestion from Reviewer cDY7, we improved our experimental performance. In addition to **Cora**, we have extended our evaluation to the larger **PubMed** dataset, as shown in R8 to Reviewer cDY7 W8. We have updated the revised manuscript with these new full-graph experimental results (with PubMed results placed in the Appendix). Furthermore, we will include experiments on synthetic graphs during the subsequent rebuttal period to comprehensively validate our theoretical claims.
>
>
> **>>>W2. Limited Model Diversity and Ablation Studies**
>
> **>>>R2:** We appreciate this suggestion. Regarding GAT and GraphSAGE, our response aligns with our discussion on Graph Transformers (R to Q1). GCNs operate on a fixed edge topology . In contrast, GAT and GraphSAGE often involve re-computing the topology (via attention mechanisms or dynamic sampling). Despite this difference, they share a fundamental mechanism: they all rely on neighborhood aggregation for information transfer. Our work analyzes the transfer error of GCNs with fixed topology on sparse graphs, providing a starting point for future research on GNNs with re-computed topology.
>
>
> **>>>Q1: Could the same convergence and transferability arguments apply to transformer-style graph networks?**
>
> **>>>R1:** We thank you for this insightful question. While our current proofs do not directly apply, our framework provides a foundational starting point.
>
> **Topological Difference:** GCNs rely on a fixed edge topology (the adjacency matrix). In contrast, Graph Transformers re-compute the topology, utilizing attention mechanisms to dynamically redefine connection weights based on node features.
>
> **Mechanistic Similarity:** Despite this difference, both architectures fundamentally rely on convolution (neighborhood aggregation) for information transfer.
>
> Our work proves that convolution on a fixed topology (GCNs) converges to a meaningful non-zero limit. This provides a starting point for future research to analyze how this convergence is maintained under dynamically "re-computed" topologies (graph transformers).
>
>
> **>>>Q2: Could you elaborate more intuitively on why stretching by the L¹-norm specifically restores a non-zero limit for sparse graphons?**
>
> **>>>R2:** We thank the reviewer for this insightful question. To explain the intuition behind $L^1$-norm stretching, we offer a physical analogy of **"restoring the correct map scale."**
>
> Classical graphons force all graphs onto a fixed $[0,1]^2$ domain. As graphs become sparse, their expected edge density ($L^1$-norm) vanishes. This is akin to compressing a vast galaxy onto a single A4 paper—the topological structure "disappears," resulting in a trivial zero limit.
>
> **The Solution (Domain Stretching):** The $L^1$-norm physically represents the "total topological information" (density). Our stretching mechanism uses this norm to **rescale the domain** (expanding the "paper" size to $\mathbb{R}_+^2$).
>
> As sparsity increases (and $L^1$-norm decreases), the stretching factor increases. This precisely counteracts the density decay, restoring the graph to its intrinsic physical scale. Consequently, the sequence converges to a stable, non-zero limit, allowing us to compare the true topological structure rather than the vanishing density.

---

> > ### Comment · Reviewer_vGFE · 2025-11-26
> >
> > Thank you for the clarifications. Although the experimental scope is still somewhat limited, the theoretical contribution remains intriguing. I will therefore maintain my original score.

---

> > > ### Author Response · Authors · 2025-11-26
> > > **Clarification on Updated PDF with Extended Experiments**
> > >
> > > We sincerely thank you for your continued engagement and for recognizing the value of our theoretical contribution.
> > >
> > > Regarding the concern about the experimental scope, we wish to respectfully note that we have **updated and uploaded the revised PDF** containing the extended experiments. We realize that we might not have emphasized this update in our previous correspondence.
> > >
> > > Specifically, in response to your suggestion, we have improved results on **Cora**, and have added results on **PubMed** and **ogbn-arxiv** (an OGB dataset) in **Section 6** and **Appendix B** (line 653).
> > >
> > > We understand your time is valuable, but if your schedule permits, we would be deeply grateful if you could briefly review these new results, which we believe broaden the experimental scope.

---

> > > > ### Comment · Reviewer_vGFE · 2025-11-26
> > > >
> > > > I have raised my score to 6 since the authors have addressed part of my concerns.

---

> > > > > ### Author Response · Authors · 2025-11-26
> > > > > **Thank You for Your Constructive Suggestion**
> > > > >
> > > > > Dear Reviewer vGFE,
> > > > >
> > > > > We are sincerely grateful for your time and the valuable suggestion. Extending experiments to diverse graphs (e.g., PubMed, OGB) has substantially strengthened our empirical scope.
> > > > >
> > > > > Thank you for helping us improve the quality of this work.
> > > > >
> > > > > Best regards,
> > > > >
> > > > > The Authors

---

### Official Review · Reviewer_zEqV · 2025-10-31

**Soundness:** 2
**Presentation:** 3
**Contribution:** 1
**Rating:** 2
**Confidence:** 4

**Summary:**

The paper studies a well known phenomena en GNNs which is the transferability of of GNNs between graphs of different sizes. In particular, this work focuses on the Stretch Graphon to be able to deal with graphs of different sparcity. All things considered, the paper is a minor extension of existing works, and the contribution (if any) is incremental.

**Strengths:**

The paper studies a relevant work which is the transferability of GNNs. I believe the paper is well written and easy to follow.

I believe the authors do make a comprehensive work citing all relevant papers.

**Weaknesses:**

I fail to understand the motivation and utility of the work. Most of the resutls are well-known and fully understood. And it seems to me that the only contribution of the work is the introduction of the Stretched graphon, which is a minor contribution to the existing graphon works. To make matters worse, the controlled sparsity is a global phenomena, and is not well suited to study graphs such as citation networks with varying degrees. The theoretical results are very difficult to interpret. And the numerical results are very noisy.

**Questions:**

- How is this work fundamentally different and not incremental from all the existing works on the topic?

- Can the authors properly explain or at least attempt to explain Theorem 1 in a way that is interpretable?

- The authors say in line 352:

Here, we guarantee transferability across graph subsequences with arbitrary sparsity, while prior
works (Keriven et al., 2020; 2021; Wang et al., 2023; 2024) analyze GCN only under fixed sparsity
settings. . In our framework, transfer errors vanish as both tm, tM and n, N increase to infinity, while
the convergence errors in Ruiz et al. (2024) converge to constants."

But this seems to be a very incremental work compared to Ruiz's previous works. How is this result fundamentally different? It seems to me that the work is incremental.

**Details Of Ethics Concerns:**

None.

---

> ### Author Response · Authors · 2025-11-20
> **Response to Reviewer zEqV (Part 1 of 2)**
>
> We appreciate your feedback and welcome this opportunity to clarify the fundamental novelty of our 'Arbitrary Sparsity' framework compared to prior works.
>
> Note: Due to the section's character limit, we split our response into two parts. **This is Part 1.**
>
> ---
>
> **>>>Response to Weakness:** We thank you for the comments. We believe these concerns stem from our core contribution and theoretical framework.
>
> 1. Concern 1: "Stretched Graphon is a minor contribution"
>
>    Our contribution is not introducing the Stretched Graphon. We explicitly state that we adopt the framework established by [1], grounded in [2, 3]. Instead, our core contribution is applying this explicit sparse model to derive the quantitative transferability bound (Theorem 1) for GCNs across arbitrary sparsity. This crucial result solves the limitations of restricted sparsity coverage [4-8] and non-vanishing errors [9] found in prior works
>
> 2. Concern 2: "controlled sparsity is not suited for varying degrees"
>
>    "Controlled sparsity" determines the **global expected edge density** (e.g., $\Theta(1/n)$), and does not mandate uniform degrees. Analyzing GCNs on sparse graphs with global sparsity has also been discussed in depth in other parallel frameworks [4-9].
>
> 3. Concern 3: "the numerical results are very noisy"
>
>    We appreciate this feedback. Adopting Reviewer cDY7's suggestion, we have improved our experimental settings. This adjustment yielded **significantly improved**, smooth, and consistent results (see R8 to Reviewer cDY7 W8). Furthermore, we extended this robust evaluation to the PubMed and Ogbn-Arxiv datasets.
>
> Our framework is grounded in foundational sparse graph theory [1-3], supports graphs with varying degrees, and covers arbitrary sparsity regimes (including the challenging $\Theta(1)$ case). With the improved full-graph experiments, our results are robust, providing strong empirical validation for our theoretical claims.
>
> ---
>
> To clearly illustrate the fundamental differences between our work and prior studies, we summarize the comparisons in the table below. **We provide the detailed discussion in Response to Q1.**
>
> **Table: Comparison of GCN Transferability Works**
>
> | Works                               | Sparsity Regime (Avg. Degree)  | Explicit Graph? | Full Convergence? (Error Vanishes to 0) |
> | :---------------------------------- | :----------------------------- | :---------------------------------: | :-------------------------------------: |
> | Ruiz et al. (2020, 2023) [11] [12]  | **Dense**                      |                 Yes                 |                   Yes                   |
> | Maskey et al. (2023) [13]           | **Dense**                     |                 Yes                 |             **Inexplicit**              |
> | Keriven et al. (2020, 2021) [4] [5] | $\Omega (\log n)$ |                 Yes                 |                   Yes                   |
> | Wang et al. (2023, 2024) [6] [7]    | $\Omega (\log n)$  |                 Yes                 |                   Yes                   |
> | Levie et al. (2021) [8]             | $\Omega (\log n)$  |               **No**                |             **Inexplicit**              |
> | Ruiz et al. (2024) [9]              | Arbitrary Sparsity             |                 Yes                 |                 **No**                  |
> | Le & Jegelka (2023) [10]            | $\Omega (1)$      |               **No**                |                   Yes                   |
> | **Ours**                            | **Arbitrary Sparsity**         |               **Yes**               |                 **Yes**                 |
>
> **Reference:**
>
> [1] Modeling sparse graph sequences and signals using generalized graphons.
>
> [2] Sparse exchangeable graphs and their limits via graphon processes.
>
> [3] Sampling perspectives on sparse exchangeable graphs.
>
> [4] Convergence and stability of graph convolutional networks on large random graphs
>
> [5] On the Universality of Graph Neural Networks on Large Random Graphs
>
> [6] Convergence of graph neural networks on relatively sparse graphs.
>
> [7] Geometric graph filters and neural networks: Limit properties and discriminability trade-offs.
>
> [8] Transferability of spectral graph convolutional neural  networks.
>
> [9] A spectral analysis of graph neural networks on dense and sparse graphs.
>
> [10] Limits, approximation and size transferability for GNNs on sparse graphs via graphops.
>
> [11] Graphon neural networks and the transferability of graph neural networks.
>
> [12] Transferability properties of graph neural networks.
>
> [13] Transferability of graph neural networks: An extended graphon approach.
>
> ---
>
> (Due to length, the response continues in the next comment. **This is Part 1 of 2.**)

---

> ### Author Response · Authors · 2025-11-20
> **Response to Reviewer zEqV (Part 2 of 2)**
>
> Note: Due to the section's character limit, we split our response into two parts. **This is Part 2.**
>
> ---
>
> **>>>Q1: Comparison with existing works.**
>
> **>>>R1**: We thank you for this question. Our work is fundamentally different, because it solves three limitations left by previous works:
>
> 1. Restricted Sparsity Regimes [4-8]. Prior works are theoretically confined to "relatively sparse" regimes ($\Theta(\log n)$ average degree). Our framework extends guarantees to **arbitrary sparsity**, covering sparser case e.g. $\Theta(1)$ average degree.
> 2. Non-vanishing Error [9]. [9] derive a bound converging to a non-zero constant . This implies that, under their framework, fully asymptotic transfer is impossible. In contrast, our Theorem 1 provides a bound guaranteed to **vanish to zero**, proving that transfer is feasible.
> 3. Implicit Modeling [10]. Other framework [10] sample operators directly. Lacking a concrete graph generative model, they cannot provide our most critical result: **physical decomposition** of transfer error into sampling error and sparsity error (governed by expected edge density).
>
> In summary, we provide a quantitative and physically interpretable transfer bound that converges to zero, solving the problem of GCN transferability under arbitrary sparsity.
>
>
> **>>>Q2: The interpretability of Theorem 1.**
>
> **>>>R2**: We thank you for this question. The interpretability of Theorem 1 is rooted in the physical meaning of our double convergence framework, which decouples sparse graph convergence into two simultaneous physical processes:
>
> 1. **Graph Size $\to \infty$** (controlled by $n$).
> 2. **Expected Edge Density $\to 0$** (controlled by $t_m \to \infty$).
>
> Based on this, Theorem 1 (Eq. 11) becomes physically interpretable by quantitatively decomposing the total transfer error into two independent sources corresponding to these processes:
>
> 1. **Sparsity Error:** The error arising from the "Expected Edge Density $\to 0$" process.
>
> 2. **Sampling Error:** The error arising from the "Graph Size $\to \infty$" process.
>
> Thus, Theorem 1 explicitly states: Transfer Error = (Convergence Error of Edge Density) + (Sampling Noise due to Finite Size). We will strengthen this physical explanation in revision.
>
>
> **>>>Q3: Comparison with Ruiz's previous works.**
>
> **>>>R3:** We thank you for this question, which allows us to articulate the fundamental difference of our work.
>
> The "convergence to constants" in Ruiz et al. (2024) [9] implies a physical limitation: even as graph size tends to infinity, GCN outputs on source and target graphs never fully converge. This indicates that asymptotic transferability is not achievable under that framework.
>
> In contrast, our Theorem 1 provides a bound guaranteed to converge to zero. Its physical meaning is that as long as the necessary conditions for sparse graph convergence are met (graph size tends to infinity and expected edge density tends to zero), GCN outputs will become consistent. This proves that asymptotic transferability is feasible.
>
> Therefore, our work is not an "incremental" improvement on the bound in [9]; it provides a fundamentally different conclusion regarding the feasibility of transfer in the sparse regime. Regarding the comparison with other works by Ruiz et al., we have provided further details in our **response to Reviewer cDY7 W4**.

---

> > ### Comment · Reviewer_zEqV · 2025-11-26
> >
> > I would like to start by thanking the authors for their thorough explanation and their work improving the numerical results of the paper. I have also read all of the reviews from the other authors.
> >
> > Unfortunately, my main comments remain unchanged. In particular, I believe that this work has very limited novelty, as even stated by the authors:
> >
> > > "We explicitly state that we adopt the framework established by [1], grounded in [2, 3]. Instead, our core contribution is applying this explicit sparse model to derive the quantitative transferability bound (Theorem 1) for GCNs across arbitrary sparsity."
> >
> > Theorem 1, is extremely similar to the works of Ruiz with a minor modification that has very limited impact in practice. Regarding the model utilized itself, it does not apply to most of the graphs seen in practice. This heavily curtails the impact of the work presented.

---

> ### Author Response · Authors · 2025-11-29
> **Response to Reviewer zEqV’s Follow-up Comments (Part 1 of 3): Summary**
>
> Note: To facilitate a quick review given the comprehensive nature of the discussion, we present a **concise overview** of our key points here in Part 1. We have placed the detailed elaboration in Part 2 and 3. **This is Part 1 of 3.**
>
> ---
>
> We appreciate the comprehensive discussion. We would like to offer the following clarifications to address your concerns.
>
> **Motivation**:  Practical scenarios like scalable GCN training and partial observation of networks inherently yield subgraphs with varying sparsity [7-9]. Analyzing GCN transferability across such arbitrary sparsity is a significant theoretical challenge not fully investigated yet.
>
> 1. **Comparison with Ruiz et al.:** Ruiz et al. [2-6] are limited by either restricted regimes (dense/"relatively sparse", e.g. Theorem 2 in [4] [5]) or non-vanishing errors (e.g. Theorem 1 in [6]). In contrast, we prove the vanishing error bound under arbitrary sparsity, a result theoretically **unattainable within the frameworks** of Ruiz et al. [2-6].
> 2. **Novelty & Contributions**: We use the model in [1] but make substantial extension in theory to quantitative GCN learning. We overcome unique technical difficulties (operator norm analysis under random stretching), e.g. Lemma 7, 8, 9 in Appendix E2, and derive explicit bounds that physically decouple transfer error into **size-driven sampling noise** and **density-driven sparsity error**, e.g. Theorem 1, 2. These quantitative physical insights are absent in prior works.
> 3. **Applicability in Practice**: Our theorem explicitly analyzes the physical impact of graph size and edge density on transfer error; this is validated by experiments on **Cora, PubMed, and ogbn-arxiv**, (shown in the updated and uploaded PDF) confirming that our theory effectively captures real-world behavior.
>
> **Reference:**
>
> [1] Modeling sparse graph sequences and signals using generalized graphons.
>
> [2] Graphon neural networks and the transferability of graph neural networks.
>
> [3] Transferability properties of graph neural networks.
>
> [4] Convergence of graph neural networks on relatively sparse graphs.
>
> [5] Geometric graph filters and neural networks: Limit properties and discriminability trade-offs.
>
> [6]  A spectral analysis of graph neural networks on dense and sparse graphs
>
> [7] GraphSAINT: Graph Sampling Based Inductive Learning Method
>
> [8] Cluster-GCN: An Efficient Algorithm for Training Deep and Large Graph Convolutional Networks
>
> [9] Sampling from Large Graphs
>
> ---
>
> Note: This concludes our summary. **If time allows, we would appreciate it if you could examine Part 2, 3**, where we provide the comprehensive details and proofs. **This part 1 of 3.**

---

> ### Author Response · Authors · 2025-11-29
> **Response to Reviewer zEqV’s Follow-up Comments (Part 2 of 3): Details**
>
> Note: Following the brief overview in Part 1, we now provide the **detailed substantiation** of our arguments. **We appreciate your time in reviewing these specific details.** This is part 2 of 3.
>
> ---
>
> **>>>Comparison with Ruiz et al.:**
>
> To clarify distinctions, we categorize the lineage of Ruiz et al.'s work based on their underlying graph models and convergence limits:
>
> 1. **Dense Graphs [2] [3] :** Ruiz et al. employ the classical graphon model which generates dense graphs, analyze the convergence from the spectral domain.
>
>    - **The "Zero Limit" Problem:** While our model also employ graphons to generate sparse graphs, sharing structural and form similarities with theirs (possibly the source of your concern), the critical difference lies in the zero graphon limit. If we simply applied Ruiz et al.’s framework to sparse graphs, the sparse graph sequence would converge to a **meaningless zero limit** due to sparsity.
>
>    - **Our Solution (Stretching):** To prevent this information loss, we employ "stretching"—physically expanding the underlying functional range (**domain size**) to compensate for the decaying edge density.
>
>    - **Overcoming New Challenges:** This stretching also introduces **stochastic range variations**, a unique difficulty that prevents the direct use of Ruiz et al.'s prior proofs. Even when sampling $G_n$ (represented by $\overline{W}_n$) from the same underlying graphon $W$, the randomness of edges causes the **stretched induced graphon $\overline{W}^{\mathfrak{s}}_n$ to fluctuate** (specifically due to domain rescaling), whereas it remains invariant in the classical formalism:
>      $$
>      \begin{aligned}
>      &\text{Classical graphon (Fixed Range):  }\\ \\ \\ \\ \\  \overline{ W } _ n \in L ^ 2([0,1] ^ 2),
>      \\\\
>      &\text{Stretched graphon (Varying Range): } \overline{ W } ^ { \mathfrak{ s } } _ n \in L ^ 2([1/ { e _ n ^ { 1/2 } }, 1/ { e _ n ^ { 1/2 } } ] ^ 2),
>      \end{aligned}
>      $$
>      where $e_n$ denotes the stochastic edge density of the random graph $G_n$. Since $e_n$ **inevitably varies** even in dense settings, it renders classical analysis frameworks invalid, as they assume a fixed domain and cannot handle such stochastic range variations. *We developed a novel operator norm analysis framework to conquer these challenges (Lemma 7, 8, 9 in Appendix E2), not only achieving the convergence rate $O(1/\sqrt{n})$ in dense case (Lemma 4 in Appendix E1) but also handling the random domain changes at the same time (Lemma 3 in Section 5.3).*
>
> 2. **Relatively Sparse Graphs [4] [5]:** These works utilize geometric graph models that are limited to dense or "relatively sparse" regimes (average degree $\Theta(n)$ or $\Theta(\log n)$). Their analysis physically relies on sufficient connection density to ensure spectral concentration (Proposition 4 in [5]). Consequently, their theory **breaks down** in the strictly sparse regimes (average degree, e.g., $\Theta(1)$) that our generalized graphon model successfully handles.
>
> 3. **Non-Vanishing Error [6]:** Ruiz et al. adopt a similar generalized graphon model. However, their derived **spectral concentration bound (Theorem 1)** is explicitly dominated by the sparsity parameter $\gamma$. This implies a physical Structural Bias: for any fixed sparse regime, a constant error remains that **does not vanish** even as graph size $n \to \infty$. Consequently, while they argue for GNNs based on expressivity, their framework fails to provide a theoretical guarantee for asymptotic transferability (i.e., convergence). In contrast, our framework proves a **vanishing transfer error** based on operator norm analysis, theoretically confirming feasibility.
>
> **Summary:** Compared to [2] and [3], we introduce "stretching" to overcome the zero limit caused by sparsity. Crucially, through our **operator norm analysis**, we derive Lemmas 7, 8, and 9 (Appendix E.2) and consolidate them into Lemma 3 (Section 5.3). This establishes a rigorous **bridge between the stretched graphon operator and the classical operator**, which constitutes one of our core technical contributions. Unlike [4] and [5] which are confined to "relative sparsity," our model is fundamentally distinct and covers **arbitrary sparsity**. Finally, in contrast to [6] where spectral analysis fails to yield full convergence in sparse settings, our operator norm-based framework successfully guarantees **vanishing error (convergence)**.
>
> ---
>
> **Note:** Due to space constraints, we continue our detailed response in the next part. Please refer to Part 3 for the discussion regarding Novelty & Contributions and Applicability in Practice. **This is part 2 of 3.**

---

> ### Author Response · Authors · 2025-11-29
> **Response to Reviewer zEqV’s Follow-up Comments (Part 3 of 3): Details**
>
> Note: Following the previous discussion, we now address the concerns regarding Novelty & Contributions and Applicability in Practice. **This is part 3 of 3.**
>
> ---
>
> **>>>Novelty & Contributions：**
>
> To model graph sequences with varying sparsity, we adopted the generalized graphon model [1] and made **substantial theoretical extensions**. The primary distinction lies in the shift from **qualitative** to **quantitative** analysis. Simply put, the difference is as follows:
> $$
> \begin{aligned}
> &\text{Qualitative analysis ([1]):  }\ \ \   \underset{ n \to \infty }{ \lim } \overline{ W }^{ \mathfrak{ s } } _ n = W^{ \mathfrak{ s } },
> \\\\
> &\text{Quantitative Error (Ours):  } \| \overline{ W }^{ \mathfrak{ s } } _ n - W^{ \mathfrak{ s } } \| _ 2 \sim O(1/n^k).
> \end{aligned}
> $$
> Evidently, our quantitative analysis provides explicit insights into the **convergence rate**. Building on this theoretical extension, we constructed the Stretched Graphon Convolutional Network (SWNN) as the limit model for GCNs in sparse regimes. We derived explicit bounds for GCN convergence and transfer errors, physically analyzing the impact of **graph size** and **edge density** on these errors, and validated these findings on real-world datasets **Cora, Pubmed, and ogbn-Arxiv**.
>
> Despite there may appear to be similarities in model architecture and conclusion forms between our work and Ruiz et al., our work is fundamentally distinct in scope and methodology. Specifically, we address the **sparsity error** driven by edge density—a component absent in Ruiz’s analysis—and resolve the **sampling error** that spectral methods cannot capture due to domain stretching. By developing a novel **operator norm analysis** (and deriving multiple new lemmas) to overcome these barriers, our work constitutes a **methodological innovation** rather than a simple incremental extension.
>
>
>
> **>>>Applicability in Practice:**
>
> For concern that the model "does not apply to most graphs seen in practice", we argue that our framework is, in fact, **more applicable** to real-world scenarios than the dense-assumption models in prior works.
>
> Real-world networks (e.g., citation graphs) typically exhibit **restricted degree growth**,  growing sublinearly, and showing significant power-law characteristics.
>
> - Classical Graphon (Ruiz et al.): Defined on a fixed $[0,1]$ domain, physically enforcing degrees to scale linearly with $\Theta (n)$ (dense). This makes it unsuitable for modeling sparse citation networks.
> - Generalized Graphon (Ours): Defined on unbounded domains. This structure naturally captures the **power-law degree distributions** and heterogeneous connectivity patterns inherent in real-world sparse graphs.
>
> [2] and [3] are limited to dense regimes ($\Theta (n)$ average degree). [4] and [5] extend only to "relatively sparse" regimes ($\Theta (\log n)$). We extend the theoretical guarantee to the **arbitrary sparsity regime**. Our framework uniquely covers the **full sublinear regimes** (e.g., $\Theta(n^\alpha)$ for $0 \leq \alpha < 1$) down to the challenging $\Theta(1)$ limit. This generalized scope ensures theoretical robustness across diverse real-world sparsity conditions.
>
> We validate this on **Cora, PubMed, and ogbn-arxiv** by explicitly controlling average degrees ($\Theta (1), \Theta (\log n), \Theta (n)$). Our updated results (partial data below, or **see uploaded PDF**) confirm our theoretical predictions:
>
> 1. **Graph Size Effect:** For a fixed edge density, transfer error diminishes as graph size increases.
> 2. **Edge Density Effect:** For a fixed graph size, lower edge density correlates with lower transfer error.
>
> Table 2: Average Transfer Errors on **PubMed Dataset  (Partial Data)**
>
> | Graph Size ($n$) | Scheme I ($\Theta (1/n)$) | Scheme II ($\Theta (\log n/n)$) | Scheme III ($\Theta (1)$) |
> | :--------------: | :-----------------------: | :-----------------------------: | :-----------------------: |
> | 100 |           0.91            |              0.78               |           0.44            |
> | 300 |           0.49            |              0.42               |           0.32            |
> | 500 |           0.36            |              0.30               |           0.26            |
> | 800 |           0.26            |              0.22               |           0.22            |
> | 1000 |           0.22            |              0.19               |           0.19            |
>
> *Note: Comprehensive results and visualizations are provided in the updated manuscript.*
>
> These results confirm that our framework effectively predicts GCN behavior in real-world sparse scenarios. We believe these results, combined with our quantitative theoretical advancements, rigorously demonstrate that our work is not a minor extension but a **necessary theoretical completion** for learning on arbitrary sparse graphs.
>
> We hope this clarification addresses your concerns.

---

### Official Review · Reviewer_cDY7 · 2025-11-02

**Soundness:** 3
**Presentation:** 2
**Contribution:** 3
**Rating:** 4
**Confidence:** 4

**Summary:**

The paper presents a novel treatment of GNN transferability across sparsity regimes. It introduces a method for generating sparse graphon sequences through truncation and rescaling and analyzes transferability bounds in this setting. The approach is conceptually clear and connects naturally with the interpretation of sparse graphs as zero-graphon limits. However, there are several weaknesses that need to be addressed prior to publication.

**Strengths:**

- Presents a novel treatment of GNN transferability **across sparsity regimes**.
- The generation of the sparse graphon sequence via truncation and rescaling is clear and intuitive, with helpful illustrations.
- The interpretation of the $L_1$-norm of the classical graphon sequence tending to zero is coherent with sparse graphs having the zero graphon as their limit.

**Weaknesses:**

- Under sparsity regimes different from those imposed in Lemmas 2–4, asymptotic convergence does not necessarily hold. As $n$ and $N$ tend to infinity, graphs do not converge unless additional assumptions are made on the corresponding graphon sequences. This is expected and not a limitation per se, but the paper slightly overstates its contribution by suggesting that transfer is possible across any two graphs associated with the same graphon regardless of such assumptions. This limitation should be highlighted, and the claims softened accordingly.

- In eqn. (7), the distance under which convergence holds should be specified. It is unclear in what sense the limits are taken, which relates to and helps explain the point above.

- I did not check the proofs in full, but I believe the constant terms related to $\mathcal{H}$ in the transferability theorem depend on the filter order $r$ and the GNN depth. This can lead to loose bounds for deeper architectures, especially given that, under weight regularization, learned weights are typically small. This is why Ruiz et al. advocate analyses leveraging the Lipschitz continuity of convolutions in the spectral domain. Those are more involved when considering polynomials over the positive reals but are well-motivated under truncation. Did you consider this perspective?

- The above also explains why the bounds in Ruiz et al. do not vanish: they can be made to vanish only if the filters’ Lipschitz constant tends to zero near small eigenvalues. Therefore, contrasting your result with Ruiz’s under this lens does not reflect an “advantage.”

- Another limitation not emphasized enough is that the tails of $X$ and $W$ must vanish; otherwise, the bound does not vanish.

- It is somewhat counterintuitive that, for fixed $n$ and $N$, the bound decreases with truncation length. Increasing truncation effectively reduces the density associated with the desirable convergence properties of graphons. Please expand on this. In particular, is this behavior an artifact of approaching the zero (trivial) graphon as truncation increases?

- The purpose of Sections 5.2 and 5.3 is unclear. They appear to introduce supporting results needed for the main theorem. If so, indicate this explicitly—specifically, that the conditions they impose are requirements for transfer, ensuring compatibility between sparsity and node growth.

- The experiments are weak. Why is performance evaluated only at node 1358 instead of over the full graph? Please extend results to the full graph. Regarding the sparsest case, where high transferability is attributed to sampling skewed toward high-degree nodes—did you check for class imbalance? A possible improvement would be to reproduce these experiments in a synthetic setting (e.g., node classification on stochastic block models) to avoid such issues and to better illustrate the theoretical findings.

- Overall, the paper should better discuss the limitations of its framework and more clearly position its contribution relative to existing work. Specifically, in the related work on transfer in sparse graphs, the authors should include
Roddenberry et al. (TSP 2023), Alimohammadi et al. (ISIT 2025), and Le and Jegelka (NeurIPS 2024).

**Questions:**

See weaknesses above.

---

> ### Author Response · Authors · 2025-11-20
> **Response to Reviewer cDY7 (Part 1 of 3)**
>
> We are grateful for your detailed and insightful review, especially for the full-graph experiment suggestion, which has significantly improved our empirical validation.
>
> Note: Due to the section's character limit, we split our response into three parts. **This is Part 1.**
>
> ---
>
> **>>>W1: Convergence of Graphs.**
>
> **>>>R1:** We thank you for this insight. The concern about the "residual error" likely stems from viewing our model as single convergence (graph size $n \to \infty$) rather than **double convergence**.
>
> Physically, sparse graph convergence involves two simultaneous processes: (1) **Graph Size $\to \infty$** (controlled by $n$); (2) **Expected Edge Density $\to 0$** (controlled by $t_m \to \infty$). Traditional models couple these; our framework decouples them to analyze arbitrary sparsity. Consequently, our theory explicitly decomposes the total error into two independent sources:
> $$ \underbrace{ \Vert \overline{W}_{m,n}^{\mathfrak{s}} - W^{\mathfrak{s}} _ { \mathbb{ R } _ + } \Vert_2 } _ {\textbf{Convergence Error} } \sim \underbrace{ \Vert \overline{ W } _ { m,n }^{\mathfrak{s}} - W^{\mathfrak{s}} _ m \Vert_2 } _ { \text{ 1.Sampling Error } \mathcal{ E }(n) } + \underbrace{ \Vert W _ m^{ \mathfrak{ s } } - W^{ \mathfrak{ s } } _ { \mathbb{ R } _ + } \Vert_2 } _ { \text{ 2.Sparsity Error } \mathcal{ E }(t_m)}, $$
> 1. **Sampling Error $\mathcal{E}(n)$:** Statistical noise caused by finite graph size $n$.
> 2. **Sparsity Error $\mathcal{E}(t_m)$:** The "residual error" you observed. This reflects the topological gap caused by finite $t_m$ (non-zero density).
>
> Thus, full convergence requires both variables to tend to infinity. Under this condition ($n, t_m \to \infty$), Theorem 1 proves the total transfer error indeed converges to zero. We will clarify this physical interpretation in the revised text.
>
>
> **>>>W2: The Limit in Eq. (7).**
>
> **>>>R2:** This question is closely related to W1.  First, the metric we use in Eq. (7) is the generalized cut distance:
> $$
> \delta _ { \Box } (W _ { \mathbb{ R } _ +,1 }, W _ { \mathbb{ R } _ +,2 })
> = \underset{ \phi _ 1, \phi _ 2 }{ \inf } \underset{ U,V \subseteq \mathbb{ R } _ + }{ \sup }
> \left| \int _ { U \times V } (W^{ \phi _ 1 } _ { \mathbb{ R } _ +,1 } - W^{ \phi _ 2 } _ { \mathbb{ R } _ +,2 }) dudv \right|,
> $$
> which is formally defined in Section 3 (Eq. 2) of our paper.
>
> Second, regarding the sense of the limits, Eq. (7) represents the double convergence framework:
> $$
> \underset{ m \to \infty }{ \lim } \underset{ n \to \infty }{ \lim } \overline{ W }^{ \mathfrak{ s } } _ { m,n }
> = \underset{ m \to \infty }{ \lim } W^{ \mathfrak{ s } } _ m
> = W^{ \mathfrak{ s } } _ { \mathbb{ R } _ + }.
> $$
>
> 1. **Inner limit ($n \to \infty$):** For a fixed density, graph $G_{m,n}$ (represented by $\overline{W}_{m,n} $) converges to its generative function $W_m$, eliminating sampling error.
> 2. **Outer limit ($t_m \to \infty$):** This sequence $W_m$ converges to the generalized graphon $W_{\mathbb{R}_+}$, eliminating the sparsity error.
>
> This structure corresponds to the physical property of sparse sequences where graph size tends to infinity while edge density tends to zero. We will clarify this convergence definition in more detail following Eq. (7) in the revision .
>
>
> **>>>W3: The Lipschitz assumption on graph filters.**
>
> **>>>R3:** We appreciate this insight. We acknowledge that the constant $C(\mathcal{ H })$ depends on network depth and filter order, which is typical for spatial domain analysis. However, our core contribution is not to seek the tightest bound in terms of depth, but to analyze GCN transferability across sparsity:
> $$
> \textbf{ Transfer Error }
> \sim
> \underbrace{ \mathcal{ E }(n) + \mathcal{ E }(N) } _ { \text{ 1.Sampling Error } }
> +
> \underbrace{ \mathcal{ E }(t _ m) + \mathcal{ E }(t _ M) } _ { \text{ 2.Sparsity Error } }.
> $$
> Our analysis successfully decomposes the total error into **graph size** and **edge density** components, proving that the error converges to zero as $n \to \infty$ and $t_m \to \infty$. While spectral analysis might yield better depth scaling, our spatial approach is sufficient to demonstrate this sparsity-driven convergence, which is the focus of our work.
>
> ---
>
> (Due to length, the response continues in the next comment. **This is Part 1 of 3.**)

---

> ### Author Response · Authors · 2025-11-20
> **Response to Reviewer cDY7 (Part 2 of 3)**
>
> Note: Due to the section's character limit, we split our response into three parts. **This is Part 2.**
>
> ---
>
> **>>>W4: Comparison with works of Ruiz et al.**
>
> **>>>R4:** We thank you for this analysis. To clarify the distinction, we categorize the development of Ruiz's related work.
>
> 1. **Dense & Relatively Sparse Graphs:** [1] [2] established transferability for dense graphs ($\Theta(n)$ degree). [3] [4] extended this to "relatively sparse" regime ($\Theta(\log n)$ degree),  cannot handle the $\Theta(1)$ case.
> 2. **Ruiz et al. (2024) "Non-Vanishing Error":** In the work we compared against [5], the spectral error bound is directly correlated with the sparsity parameter $\gamma$. Thus, for any sparse graph ($\gamma > 0$), a fixed structural bias remains that does not vanish.
>
> The view that "non-vanishing error stems solely from filters' Lipschitz constants", aligns with earlier dense graph analyses [2]. In the sparse model of [5], the error is structural. In contrast, our double convergence framework solves transferability with arbitrary sparsity. As long as graph size increases ($n \to \infty$) and edge density decreases ($t_m \to \infty$), our transfer error vanishes, providing a guarantee of asymptotic consistency that prior works do not.
>
>
> **>>>W5: How the bound vanishes.**
>
> **>>>R5:** We thank you for this keen observation. You are entirely correct: the truncation tails of $W _ { \mathbb{ R } _ + }$ and $X _ { \mathbb{ R } _ + }$ must vanish. for the total error bound to vanish. This is not an overlooked limitation but a core feature of the sparse graph model we adopted [6]. Physically, sparse graph convergence involves two simultaneous processes:
>
> 1. Graph size tends to infinity (controlled by $n$);
> 2. Expected edge density tends to zero (controlled by $t_m \to \infty$).
>
> **This requirement (tails must vanish) is an explicit prerequisite of our model, as it describes the decreasing edge density.**
>
>
> **>>>W6: Physical Meaning of Fixed $n$, Varying $t_m$**.
>
> **>>>R6:** We appreciate this question regarding the physical implications of the bounds. As described in R1 to W1, our framework decouples graph size and edge density: $n$ independently controls the graph size, $t_m$ independently controls the expected edge density.
>
> The scenario of fixing $n$ while varying $t_m$ implies analyzing a graph of fixed size across **different sparsity regimes**:
> $$G _ { t _ { m _ 1 }, n } \sim \Theta( \log n/n )\  \to \ G _ { t _ { m _ 2 }, n } \sim \Theta(1/n).$$
> This decoupling capability is precisely what enables us to establish transferability across arbitrary sparsity.
>
>
> **>>>W7: The purpose of Sections 5.2 and 5.3.**
>
> **>>>R7:** You are correct: Sections 5.2 and 5.3 provide the core supporting results for Theorem 1. Our double convergence framework [6] decouples sparse graph convergence into two physical processes: Graph Size ($n$) and Expected Edge Density ($t_m$) . Consequently, these sections quantify the two decoupled error sources:
>
> **Section 5.3** quantifies sampling error (noise due to finite $n$).
>
> **Section 5.2** quantifies sparsity error (topological gap due to finite $t_m$).
>
> We will add a detailed clarification statement in Section 5 to make this structure explicit.
>
> ---
>
> (Due to length, the response continues in the next comment. **This is Part 2 of 3.**)

---

> ### Author Response · Authors · 2025-11-20
> **Response to Reviewer cDY7 (Part 3 of 3)**
>
> Note: Due to the section's character limit, we split our response into three parts. **This is Part 3.**
>
> ---
>
> **>>>W8: Evaluating performance of the full graph.**
>
> **>>>R8:** We greatly appreciate this suggestion. Adopting it, we redesigned our experiments to measure the average transfer error over the full sampled graph.
>
> This adjustment significantly improved results on **Cora**, yielding smoother curves and clearer distinctions between sparsity schemes.  Furthermore, we extended this robust evaluation to the larger **PubMed** dataset and **Ogbn-Arxiv** dataset. Results (see partial data below) are highly consistent, confirming that error converges as graph size increases ($n \to \infty$) and ***edge density decreases (Schemes I $\to$ III)***. We have updated Section 6 and the Appendix with these comprehensive results **in the revised PDF**.
>
> Table 1: Average Transfer Errors on **Cora Dataset (Partial Data)**
>
> | Graph Size ($n$) | Scheme I ($\Theta (1/n)$) | Scheme II ($\Theta(\log n/n)$) | Scheme III ($\Theta(1)$) |
> | :--------------: | :-----------------------: | :----------------------------: | :----------------------: |
> |       100        |           1.37            |              1.13              |           0.81           |
> |       200        |           0.86            |              0.74              |           0.66           |
> |       300        |           0.61            |              0.58              |           0.57           |
> |       400        |           0.50            |              0.48              |           0.48           |
> |       500        |           0.43            |              0.43              |           0.43           |
>
> Table 2: Average Transfer Errors on **PubMed Dataset  (Partial Data)**
>
> | Graph Size ($n$) | Scheme I ($\Theta (1/n)$) | Scheme II ($\Theta (\log n/n)$) | Scheme III ($\Theta (1)$) |
> | :--------------: | :-----------------------: | :-----------------------------: | :-----------------------: |
> |       100        |           0.91            |              0.78               |           0.44            |
> |       300        |           0.49            |              0.42               |           0.32            |
> |       500        |           0.36            |              0.30               |           0.26            |
> |       800        |           0.26            |              0.22               |           0.22            |
> |       1000       |           0.22            |              0.19               |           0.19            |
>
> *Note: Comprehensive results and visualizations are provided in the updated manuscript.*
>
>
> **>>>W9: Related Works.**
>
> **>>>R9:** We thank you for these recommendations.
>
> 1. [7] explores graph topology inference tasks based on the idea that similar motif densities lead to similar expected filter outputs.
> 2. [8] analyzes the sampled training of GNNs, proposing that GNN parameters learned on subgraphs are close to the optimal parameters learned on the full graph.
> 3. [9] proposes a transfer framework based on graphops (continuous operators), exploring the transfer performance of operators obtained through direct discretization.
>
> [7] [8] focus on graph topology inference and training convergence tasks, respectively.  In contrast, our work addresses transferability: analyzing the output difference of GCNs with fixed parameters across sparse graphs. [9] does not generate concrete graphs. Unlike our generative model, this approach cannot physically decompose the transfer error into sampling error and sparsity error. We have added a discussion in Section 2.
>
>
> **Reference:**
>
> [1] Ruiz et al. (2020). Graphon neural networks and the transferability of graph neural networks.
>
> [2] Ruiz et al. (2023). Transferability properties of graph neural networks.
>
> [3] Wang et al. (2022). Convergence of graph neural networks on relatively sparse graphs.
>
> [4] Wang et al. (2023). Geometric graph filters and neural networks: Limit properties and discriminability trade-offs.
>
> [5] Ruiz et al. (2024). A spectral analysis of graph neural networks on dense and sparse graphs.
>
> [6] Ji et al. (2024). Modeling sparse graph sequences and signals  using generalized graphons.
>
> [7] Roddenberry et al. (TSP 2023). Enhanced Graph-Learning Schemes Driven by  Similar Distributions of Motifs.
>
> [8] Alimohammadi et al. (ISIT 2025). A Local Graph Limits Perspective on  Sampling-Based GNNs.
>
> [9] Le & Jegelka (2023). Limits, approximation and size transferability for GNNs on sparse graphs via graphops.

---

### Official Review · Reviewer_VfQf · 2025-11-04

**Soundness:** 3
**Presentation:** 3
**Contribution:** 2
**Rating:** 4
**Confidence:** 4

**Summary:**

The paper looks at transferability of graph convolutional networks, focusing on graphs with varying degree of sparsity. They introduced stretched Graphon Convolutional Networks (sWNN) based on extended graphons.

**Strengths:**

The paper's math is correct as far as the reviewer can check, with good illustrations and is relatively easy to follow. Overall, the technical contribution of the paper is substantial. The approach used in the paper can deal with very sparse graphs, which is both practical and theoretically challenging.

**Weaknesses:**

The paper largest weaknesses are its motivation highlighted by certain key misconceptions:
1. Sparsity and big-O notation: when discussing sparsity of graphs in the literature, most authors (that are cited in the paper) refer to sparsity of a **graph sequence** and therefore wrote the level in asymptotic notations (e.g. $\Theta(\log n)$ average degree). This means that all of their results are valid for graph sequences whose degrees are constant factors away from one another. This include Keriven, Bietti and Vaiter's "Convergence and stability of graph convolutional networks on large random graphs" and "On the universality of graph neural networks on large random graphs"; as well as Wang, Ruiz and Ribeiro's "Convergence of graph neural networks on relatively sparse graphs" and "Geometric graph filters and neural networks: Limit properties and discriminability trade-offs". These works were cited repeatedly to motivate the current paper's application to various sparse graph regimes (e.g. line 353  "while prior works (Keriven et al., 2020; 2021; Wang et al., 2023; 2024) analyze GCN only under fixed sparsity settings."), which is incorrect. By definition, a function g being in $\Theta(f(n))$ simply means that there are constants c and C such that there exists a N, where $cf(n) \leq g(n) \leq Cf(n)$ for all n > N.
2. Generalized graphon (Definition 1) are known in the literature as (a part of) graphex (e.g. Definition 4 in Borgs, Chayes, Dhara and Sen's "Limit of sparse configuration models and beyond: graphexes and multigraphexes). This is minor, but is a crucial literature to study (and cite).
3. Missing key comparisons to unbounded graphon/operator. The two papers that closely implement the authors idea of 'stretching' a graphon is Maskey, Levie and Kutyniok's "Transferability of graph neural networks: an extended graphon approach" and Levie et al. (2021),  "Transferability of spectral graph convolutional neural networks" where they use unbounded operators to model very sparse graphs (even O(1) average degree). Compared to the approach of the paper, which changes the domain of a graphon to being unbounded, unbounded operator changes the range to being unbounded. The two are equivalent if there is a change of measure from one to another (which is probably the vast majority of cases). Seeing that the unbounded operator approach can do any and all things that the paper is claiming to do, a comprehensive comparison is warranted.

I am willing to raise my score if my concerns are fully addressed in the rebuttal.

**Questions:**

See Weaknesses.

---

> ### Author Response · Authors · 2025-11-20
> **Response to Reviewer VfQf**
>
> We sincerely thank you for the feedback, which helps us clarify our positioning regarding 'fixed sparsity' and the comparison with unbounded operators.
>
> **>>>W1: Sparsity and big-O notation**
>
> **>>>R1:** We thank you for the clarification regarding asymptotic notations.  We acknowledge that prior results are valid for sequences within specific regimes. We wish to clarify our true intention: prior works are confined to **restricted sparsity regimes**.
>
> 1. Prior Limits: While prior works [1-4] analyze sparse regimes, $\Theta(\log n)$ average degree represents their **sparsest case**. Sparser graphs (e.g., $\Theta(1)$ average degree) are not covered.
> 2. Our Advantage: Our framework uses a generalized sparse model without such restrictions. It covers **arbitrary sparsity**, extending to the challenging $\Theta(1)$ case and even sparser regimes, as validated by our Scheme I experiments.
>
> We model cross-sparsity transfer not only within a specific sparsity regime (e.g., $\Theta(\log n)$), but also across different regimes (e.g., between $\Theta(\log n)$ and $\Theta(1)$). To reflect greater precision, we adopt $\Omega$ to denote the sparsity coverage range and revise the text to clarify that prior works "...analyze GCN only within restricted sparsity regimes".
>
>
> **>>>W2: Related Literature**.
>
> **>>>R2:** We thank you for recommending Borgs et al. [7]. We clarify that our chosen framework [8] and the graphex framework [7] are parallel formalisms that both originate from the foundational works of [5] [6]. While graphex focuses on structural limits, [8] provides a signal processing perspective suitable for our analysis. We add the citation and supplement the discussion in Section 3.
>
>
> **>>>W3: Comparison with unbounded operators.**
>
> **>>>R3:** We thank you for the suggestion. While mathematically similar (unbounded), these works [9] [10] differ fundamentally from ours in sparsity scope and graph modeling:
>
> 1. **Sparsity Scope:** Maskey et al. [9] focus on dense graphs, Levie et al. [10] are confined to relative sparsity ($\Theta(\log n)$ average degree), as noted in [4] [5] [11]. In contrast, our framework covers arbitrary sparsity, including sparser case (e.g., $\Theta(1)$ average degree), which we validate experimentally (Scheme I).
> 2. **Graph Modeling:** These works [9, 10] analyze operators without generating concrete graphs. Our generative model enables us to physically decompose the transfer error into expected edge density and graph size components (Theorem 1), providing interpretability that implicit methods lack.
>
> The "unbounded value" in [9] [10] **arises from directly sampling operators, rather than being constructed based on sparsity**. In contrast, our framework employs stretching to offset sparsity, thus offering a wider scope of adaptability.
>
>
> **Reference:**
>
> [1] Keriven et al. (2020). Convergence and stability of graph convolutional networks on large random graphs
>
> [2] Keriven et al. (2021). On the Universality of Graph Neural Networks on Large Random Graphs
>
> [3] Wang et al. (2022). Convergence of graph neural networks on relatively sparse graphs.
>
> [4] Wang et al. (2023). Geometric graph filters and neural networks: Limit properties and discriminability trade-offs.
>
> [5] Borgs et al. (2018). Sparse exchangeable graphs and their limits via graphon processes.
>
> [6] Borgs et al. (2019). Sampling perspectives on sparse exchangeable graphs.
>
> [7] Borgs et al. (2021). Limits of sparse configuration models and beyond.
>
> [8] Ji et al. (2024). Modeling sparse graph sequences and signals  using generalized graphons.
>
> [9] Maskey et al. (2023). Transferability of graph neural networks: An extended graphon approach.
>
> [10] Levie et al. (2021). Transferability of spectral graph convolutional neural  networks.
>
> [11] Le & Jegelka, (2023).  Limits, approximation and size transferability for gnns on sparse graphs via graphops.

---

> > ### Comment · Reviewer_VfQf · 2025-11-26
> >
> > I thank the authors for the thoughtful reply. I'm raising my score since the authors have addressed my concerns in that the significance of their contributions lie in the ability to analyze a sequence of graphs that is discontinuous in sparsity/average degree for the corresponding limit framework.

---

> > > ### Author Response · Authors · 2025-11-26
> > > **Thank You for Your Acknowledgment**
> > >
> > > Dear Reviewer VfQf,
> > >
> > > We are sincerely grateful for your reply and acknowledgement. We are particularly encouraged that you precisely identified the core significance of our work: the ability to analyze graph sequences that are discontinuous in sparsity/average degree.
> > >
> > > Thank you once again for your time and valuable feedback.
> > >
> > > Best regards,
> > >
> > > The Authors

---

### Author Response · Authors · 2025-12-02
**General Response: Status Update, Score Increase, and Key Clarifications**

Dear Reviewers, ACs, and SACs,

We sincerely thank you for your dedication and constructive discussions. We especially appreciate the AC's additional time and effort given the recent circumstances. To facilitate the review, we provide a concise status update:

- **Revised PDF:** Updated with experimental expansions (**Cora, PubMed, ogbn-arxiv**) and theoretical clarifications.
- **Score Increases:** On ***Nov 26 02:43 (AOE)*** and ***Nov 26 03:54 (AOE)*** before the system bug, **Reviewer VfQf** and **Reviewer vGFE** confirmed their concerns were resolved and **raised their scores to 6**.
- **Ongoing Discussions:** **Reviewer zEqV** posted follow-up questions just before the interruption. Our responses are summarized below.

**Summary of Responses:** Reviewer VfQf **acknowledged the theoretical significance** of our arbitrary sparsity analysis, and Reviewer vGFE **verified** that the new full-graph experiments resolved the concerns. Reviewer cDY7's **constructive suggestions** drove a major upgrade in our experiments. Regarding Reviewer zEqV, we addressed the **conceptual misalignment** concerning sparsity and stretching, clarifying why prior analysis methods **cannot be simply generalized** to our arbitrary sparsity setting.

We hope these clarifications help address the reviewers' concerns and facilitate the final assessment by the ACs and SACs.

*(Note: Below is a concise summary of the key points from our rebuttal.)*

---

**Reviewer VfQf  (Positive - Score Raised - (AOE) Nov 26 02:43)**

- **Sparsity Regime (W1):** Our framework is unique to handle arbitrary/discontinuous sparsity, compared to sparsity-restricted prior works.
- **Unbounded Range (W3):** The "unbounded range" (via stretching) overcomes the "zero limit" issue caused by sparsity, while "unbounded operators" in prior works do not consider sparsity.

**Reviewer vGFE (Positive - Score Raised - (AOE) Nov 26 03:54)**

- **Experimental Scope (W1):** Concerns resolved by adding **PubMed and ogbn-arxiv** results.
- **Extension to GNNs (W2, Q1):** Our fixed-topology GCN analysis serves as the foundation for GNNs with dynamic topologies (e.g., Graph Transformers).

**Reviewer cDY7  (Constructive - Discussion Pending)**

- **[Major Update] Experiments (W8):** We adopted the suggestion of full-graph evaluation to eliminate noise and expanded to **PubMed and ogbn-arxiv**, confirming consistent convergence.
- **Theory & Convergence (W1-W6):** We clarified that our framework physically decouples graph size and density, and our method guarantees full convergence under **arbitrary sparsity**.

**Reviewer zEqV (Discussion Ongoing)**

- **Differentiation (Q1, W1):** We distinguished our work from prior works by resolving restricted sparsity and lack of interpretability. We clarified "Stretch" is a critical tool to prevent the "zero limit" issue.

- **Degree Distribution (W2):** "Controlled sparsity" accommodates **power-law distributions** (e.g., citation graphs) and does not mandate uniform degrees.

- **[Detailed Clarification on Follow-up Comments by Reviewer zEqV]:** We clarified that prior analysis methods (dense/restricted graphs) **cannot be simply generalized** to our setting due to fundamental mathematical barriers.

  - **Fundamental Distinction:** Classical graphons operate on a *fixed range*, whereas stretched graphons require *varying random ranges*. Results from the former are mathematically inapplicable to the latter.
  - **Theoretical Advancement:** We advanced the field from *qualitative* analysis to **quantitative bounds**, providing explicit convergence rates previously missing.
  - **Validation:** Our theory accurately predicts error decay (as size $\uparrow$ or density $\downarrow$), behavior now robustly validated by updated experiments on **Cora, PubMed, and ogbn-arxiv.**


  (*Note: For full details, please refer to our **three consecutive responses**--posted immediately after the reviewer's follow-up comment--under the* ***3rd Official Review thread (Reviewer zEqV).***)

---

We once again express our gratitude to the AC and all reviewers for their significant efforts, especially given the challenging timeline. We believe that the constructive feedback has substantially strengthened our work, and we sincerely hope that these revisions have satisfactorily addressed the concerns.

Sincerely,

The Authors of Submission 11617

---

### Meta-Review · Area_Chair_Ahim · 2025-12-29

**Summary:**

In this paper, the authors study the GCN transferability bound for size generalization. Specifically, the authors introduce Stretched Graphon Convolutional Networks (SWNNs) based on the graphon model, aiming to overcome the limitation of restricted sparsity in previous studies. Experimental results validate the theoretical findings.

Initially, the reviewers raised various valid concerns regarding the theoretical contributions, including the novelty compared to the literature, applicability, and technical details, as well as insufficient experimental support, as reflected by the rather negative initial scores. After the rebuttal, two reviewers are likely to increase the score, and another reviewer participates actively in the discussion. The updated score makes the paper a borderline case. Though the theoretical contributions are potential contributions, the scope and whether the results are significant for the general audience are somewhat limited, as reflected by none of the reviewers particularly championing the paper. Considering the highly competitive nature of ICLR (at least in my batch), I vote for rejection, but encourage the authors to further revise their paper for another venue.

**Reviewer Concerns:**

Reviewer VfQf:
W1-Sparsity and big-O notation: partially addressed.
W2-Generalized graphon... : partially addressed.
W3-Missing key comparisons to unbounded graphon/operator: partially addressed.

Reviewer cDY7:
The reviewer raised several rather technical concerns, and I can only guess that they are partially addressed.

Reviewer zEqV:
W1-How is this work fundamentally different: partially addressed.
W2-Theorem 1: partially addressed.
W3-Comparison with Ruiz's previous works: partially addressed.

Reviewer vGFE:
W1-Experimental Scope Limitations: partially addressed.
W2-Limited Model Diversity and Ablation Studies: partially addressed.

**Reviewer Scores:**

For Reviewer VfQf, the initial rating is 4, and it is likely to increase to 6.

For Reviewer cDY7, the initial rating is 4, and it is likely to stay at 4 or increase to 6.

For Reviewer zEqV, the initial rating is 2, and it is likely to stay at 2 or increase to 4.

For Reviewer vGFE, the initial rating is 4, and it is likely to increase to 6.

---

### Decision · Program_Chairs · 2026-01-26

Reject